# PHF19 mediated regulation of proliferation and invasiveness in prostate cancer cells

Payal Jain[1†], Cecilia Ballare[1†], Enrique Blanco[1], Pedro Vizan[1], Luciano Di Croce[1,2,3]*

[1]Centre for Genomic Regulation (CRG), The Barcelona Institute of Science and Technology, Barcelona, Spain; [2]Universitat Pompeu Fabra (UPF), Barcelona, Spain; [3]ICREA, Barcelona, Spain

**Abstract** The Polycomb-like protein PHF19/PCL3 associates with PRC2 and mediates its recruitment to chromatin in embryonic stem cells. PHF19 is also overexpressed in many cancers. However, neither PHF19 targets nor misregulated pathways involving PHF19 are known. Here, we investigate the role of PHF19 in prostate cancer cells. We find that PHF19 interacts with PRC2 and binds to PRC2 targets on chromatin. PHF19 target genes are involved in proliferation, differentiation, angiogenesis, and extracellular matrix organization. Depletion of PHF19 triggers an increase in MTF2/PCL2 chromatin recruitment, with a genome-wide gain in PRC2 occupancy and H3K27me3 deposition. Transcriptome analysis shows that PHF19 loss promotes deregulation of key genes involved in growth, metastasis, invasion, and of factors that stimulate blood vessels formation. Consistent with this, *PHF19* silencing reduces cell proliferation, while promotes invasive growth and angiogenesis. Our findings reveal a role for PHF19 in controlling the balance between cell proliferation and invasiveness in prostate cancer.

*For correspondence:
luciano.dicroce@crg.es

†These authors contributed equally to this work

Competing interests: The authors declare that no competing interests exist.

## Introduction

Polycomb group (PcG) proteins are transcriptional regulators involved in embryonic development, cell differentiation, and maintenance of cell identity. Deregulation of PcG has been linked to anomalous activation of differentiation pathways, carcinogenesis and cancer progression (*Margueron and Reinberg, 2011*; *Schuettengruber et al., 2017*; *Pasini and Di Croce, 2016*). PcG proteins form two major Polycomb repressive complexes (PRC): PRC1, responsible for the deposition of H2AK119ub1, and PRC2, which catalyzes H3K27 methylation (*Di Croce and Helin, 2013*). The PRC2 core, formed by EZH1/2, SUZ12, EED, and RBBP4/7, can interact with additional accessory proteins that modulate its function; these include Polycomb-like proteins (*Vizán et al., 2015*). The *Polycomb-like* (*Pcl*) gene was originally discovered in *Drosophila melanogaster* and presented the same mutant phenotypes as the Polycomb genes (*Duncan, 1982*). Three mammalian paralogs of *Drosophila Pcl*, termed *PHF1/PCL1*, *MTF2/PCL2*, and *PHF19/PCL3*, have been characterized to date, mainly in the context of mouse embryonic stem cells (ESCs) (*Cao et al., 2008a*; *Sarma et al., 2008*; *Casanova et al., 2011*; *Walker et al., 2010*; *Ballaré et al., 2012*; *Brien et al., 2012*). PHF19 plays a pivotal role in gene silencing through its ability to recognize the epigenetic mark H3K36me3 on active genes *via* its Tudor domain, and mediate PRC2 recruitment (*Ballaré et al., 2012*; *Brien et al., 2012*). Similar properties were later reported for the other members of the PCL family (*Cai et al., 2013*; *Li et al., 2017*). The above-mentioned studies extensively describe these mechanisms for ESCs, in which silencing of lineage-specific genes is essential to maintain pluripotency.

In humans, *PHF19* encodes a long (PHF19L) and a short (PHF19S) isoform, that are generated by alternative splicing and are both overexpressed in a wide variety of cancers (*Wang et al., 2004*;

*Boulay et al., 2011*). PHF19 interacts with the tumor suppressor HIC1 and thus mediates PRC2 recruitment to a subset of HIC1 target genes (*Boulay et al., 2012*). Further, through the induction of PHF19, p-Akt has been reported to promote melanoma progression, (*Ghislin et al., 2012*). In addition, PHF19 can promote proliferation in hepatocellular carcinoma, glioma, and ovarian cancers (*Xu et al., 2015*; *Lu et al., 2018*; *Tao et al., 2018*) and can induce glioblastoma progression, mediated by β-catenin (*Deng et al., 2018*). However, despite these efforts to understand the role of PHF19 in different cancer models, a comprehensive analysis that identifies the genetic targets and pathways controlled by PHF19 has so far not been reported.

Enhancer of Zeste 2 (EZH2), the enzymatic component of PRC2 that methylates of lysine 27 at histone H3, is often overexpressed in prostate cancer (*Koh et al., 2011*; *Bracken, 2003*; *Varambally et al., 2002*). EZH2 overexpression is associated with the acquisition of new PRC2 targets, including tumor suppressors, and with poor outcome in disease (*Cao et al., 2008b*; *Shin and Kim, 2012*; *Wu et al., 2014*; *Wee et al., 2014*; *Ding et al., 2014*). In addition, cooperation of EZH2 with the androgen receptor and with DNA methyltransferases can reinforce PRC2 mediated-silencing at target genes (*Zhao et al., 2012*; *Moison et al., 2013*; *Moison et al., 2014*). Further, an oncogenic function of EZH2 in prostate cancer, independent of its role as a transcriptional repressor, was also reported. This involves the ability of EZH2 to switch from a Polycomb repressor to a co-activator for critical transcription factors including the androgen receptor (*Xu et al., 2012*). Whether or how PHF19 modulates the function and targets of the EZH2 in prostate cancer remains to be explored.

In this study, we report a novel role for PHF19 in controlling the balance between growth and invasiveness in prostate cancer. We show that PHF19 interacts with PRC2, and that both co-localize at chromatin. Depletion of PHF19 causes upregulated MTF2/PCL2 expression and increased MTF2 recruitment to chromatin, along with a genome-wide gain in PRC2 occupancy and increased H3K27me3 deposition. This in turn leads to transcriptional deregulation of key genes involved in the control of proliferation, angiogenesis, metastasis, and invasion. Finally, with the loss of PHF19, prostate cancer cells switch to a less proliferative but more aggressive phenotype.

## Results

### PHF19L interacts with the PRC2 complex in prostate cancer cells

Two isoforms of PHF19 are generated in humans: PHF19L has a Tudor domain, two PHD fingers, an extended homology (EH) domain, and a chromo-like domain, while PHF19S, contains only the N-terminal Tudor and PHD1 domains (*Figure 1A*). To investigate the role of PHF19 in prostate cancer, we first evaluated its expression in two common human prostate cancer cell models, the poorly-differentiated metastatic PC3 and DU145 cell lines, as well as in a normal counterpart, the prostate epithelial cell line RWPE1. PHF19L/S were both expressed in PC3 and DU145 cells, and at higher levels as compared to those in RWPE1 cells (*Figure 1B*).

We then analyzed protein interactors for each PHF19 isoform by mass spectrometry (MS). To this end, PC3 cells stably expressing a FLAG-tagged version of PHF19L or PHF19S, or a FLAG-tagged empty vector (as a control) were generated, and subjected to FLAG affinity purification followed by MS. As previously reported for ESCs (*Ballaré et al., 2012*; *Brien et al., 2012*), PHF19L mainly interacted with the core subunits of the PRC2 complex. On the other hand, PHF19S did not interact with any PRC2 component (*Figure 1C*). This suggests that PHF19L and PHF19S have different functions in prostate cancer. Co-immunoprecipitation assays confirmed that only FLAG-PHF19L interacted with EZH2 (*Figure 1—figure supplement 1A*). We then validated the interaction of PHF19L with the PRC2 complex at endogenous level. Indeed, PHF19L co-immunoprecipitated with EZH2 and SUZ12 in PC3 and DU145 cells (*Figure 1D,E*). Depletion of PHF19L with specific short hairpin RNAs (shRNAs) disrupted these interactions but did not affect the stability of the PRC2 complex, as the association between the core subunits EZH2 and SUZ12 remained unchanged (*Figure 1D,E*). In addition, by using specific knockdowns for each PHF19 isoform, we confirmed that the depletion of one of the isoforms had no impact on the expression of the other (*Figure 1—figure supplement 1B*).

### PHF19L co-localizes with PRC2 on chromatin

We next investigated the cellular localization of PHF19 in DU145 and PC3 cells. Cell fractionation revealed that, in both prostate cancer cell lines, PHF19L is mainly present on chromatin, whereas

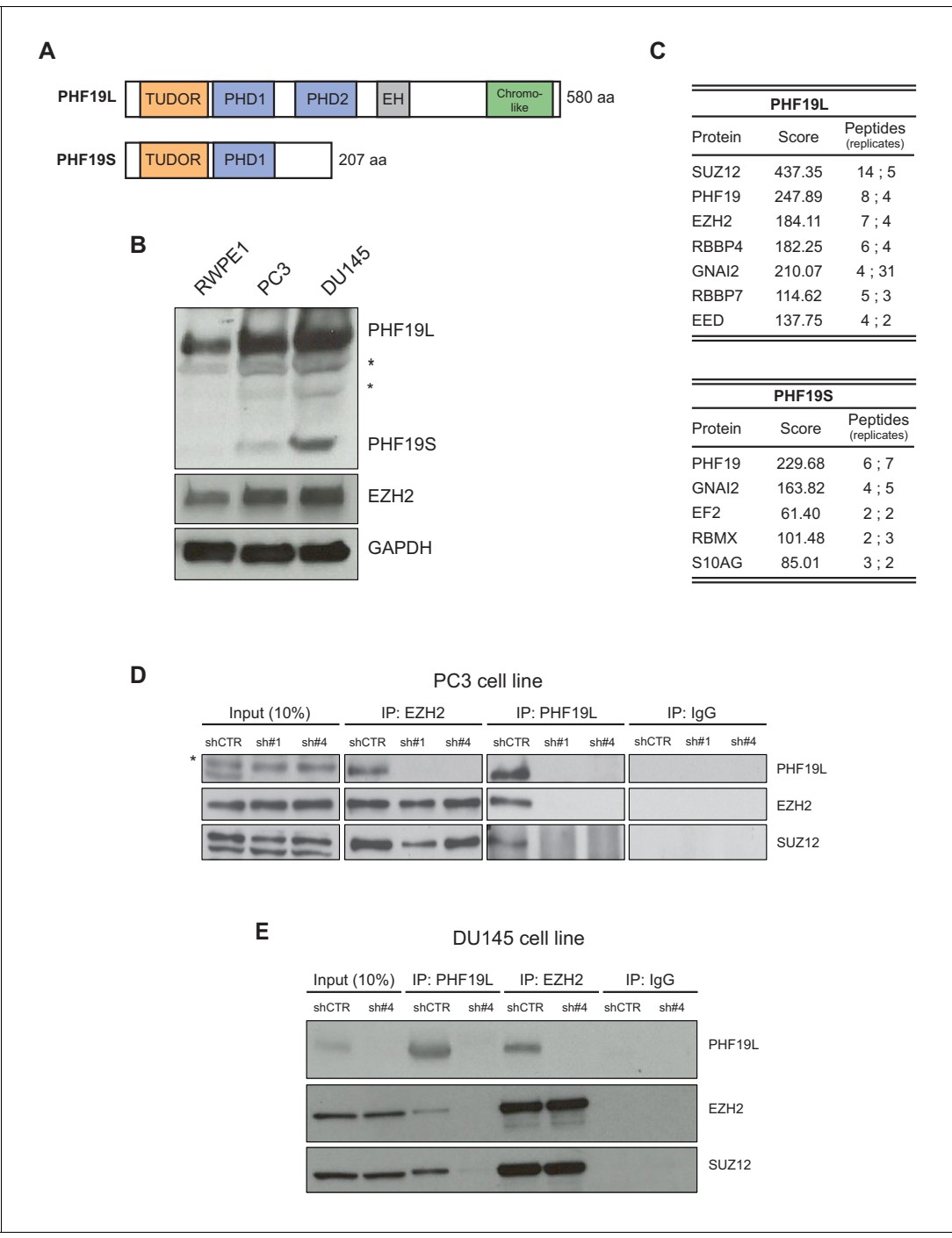

**Figure 1.** PHF19L associates with PRC2 in prostate cancer cells. (**A**) Schematic representation of PHF19L and PHF19S and their domains. (**B**) Western blot analysis showing expression of PHF19L, PHF19S, EZH2, and GAPDH in RWPE1, PC3, and DU145 cells. *, non-specific bands (**C**) Summary of the main interactors of PHF19L and PHF19S identified by mass spectrometry (MS). PC3 cells stably expressing FLAG-tagged PHF19L or PHF19S, or FLAG-tagged empty vector (control), were subjected to FLAG affinity purification followed by MS. The table displays the score and the peptide count from two independent experiments. (**D,E**) Endogenous co-immunoprecipitation (IP) of PHF19L with EZH2 or SUZ12 in control (shCTR) and PHF19L-depleted (shPHF19L#1 or shPHF19L#4) PC3 cells (**D**) or DU145 cells (**E**). IgG was used as a control. *, non-specific band.

The online version of this article includes the following figure supplement(s) for figure 1:

**Figure supplement 1.** PHF19L associates with PRC2 in prostate cancer cells.

PHF19S is cytoplasmic (*Figure 2A*). Moreover, ectopically overexpressed PHF19S in DU145 cells was also only present in the cytoplasm, indicating that the lack of signal of PHF19S on chromatin was not due to technical limitations in detecting low levels of protein (*Figure 2—figure supplement 1A*).

In order to explore the role of PHF19L on chromatin in prostate cancer, we carried out chromatin immunoprecipitation using an anti-PHF19 antibody followed by high-throughput sequencing (ChIP-seq) in DU145 cells. To ensure the specificity of the detected sites, ChIP-seq assay was also performed in PHF19L-depleted (shPHF19L#4) cells. We performed two independent biological ChIP-seq replicates to assess the statistical significance of the results. Differential binding analysis (Diff-Bind) (*Ross-Innes et al., 2012*) identified a total of 1245 significant PHF19-binding regions (peaks, P value < 0.05 and FDR < 0.2), corresponding to 1010 target genes (*Supplementary file 1*, *Figure 2—figure supplement 1B*). PHF19 was mainly bound near the transcription start site (TSS) of the target genes (*Figure 2B,C* and *Figure 2—figure supplement 1C*). The ChIP signal was strongly reduced upon knockdown of PHF19L (*Figure 2B* and *Figure 2—figure supplement 1D,E*). The presence of PHF19L was further validated in a subset of target genes by ChIP-qPCR and the specificity of the signal was confirmed by using two different shRNAs (shPHF19L#4 and shPHF19L#B) (*Figure 2—figure supplement 1F*).

Gene ontology (GO) analysis revealed a significant enrichment of PHF19L at genes involved in essential biological processes, such as regulation of cell development and differentiation, cell proliferation, various signaling pathways, and extracellular matrix organization, highlighting its potential role in prostate cancer (*Figure 2D*).

To investigate whether PHF19L co-localizes with the PRC2 on chromatin, we performed ChIP-seq experiments for EZH2, SUZ12, and H3K27me3 in DU145 cells (shCTR). In concordance with the mass-spectrometry data showing that PHF19L interacts with PRC2, PHF19L target genes were also occupied by EZH2 and SUZ12 (*Figure 2E*, upper panels). Additionally, a strong correlation was found between signals of PHF19 with EZH2, as well as with SUZ12, in the PHF19L ChIP-seq peaks (*Figure 2E*, lower panels). These results were confirmed in a second set of ChIP-seq replicates (*Figure 2—figure supplement 1G*). Consistently, PHF19L target genes were also strongly enriched in H3K27me3, and a significant correlation was observed between PHF19 and H3K27me3 signals in PHF19L peaks (Replicate 1, *Figure 2F* and Replicate 2, *Figure 2—figure supplement 1H*). In fact, almost all of the PHF19L target genes (93%) were decorated by H3K27me3 (*Figure 2—figure supplement 1I*).

## Genome-wide increase in PRC2 in the absence of PHF19L

In ESCs, PHF19 is required for stable association of PRC2 at target genes (*Ballaré et al., 2012*; *Brien et al., 2012*). To investigate whether PHF19 also affects PRC2 binding in prostate cancer, we analyzed the genome-wide occupancy of PRC2 subunits and the H3K27me3 mark, in control (shCTR) and PHF19L-depleted (shPHF19L#4) DU145 cells. Unexpectedly, loss of PHF19L did not reduce but rather triggered a global increase in the recruitment of EZH2 and SUZ12, as well as in H3K27me3 deposition (*Figure 3A*). This observation was confirmed in a second set of ChIP-seq replicates (*Figure 3—figure supplement 1A*). Indeed, in the absence of PHF19L, there was a substantial increase of EZH2, SUZ12, and H3K27me3 ChIP-seq peaks and target genes (*Supplementary file 1*, *Figure 3B* and *Figure 3—figure supplement 1B*). Furthermore, differential binding analysis (Diff-Bind) found a significant gain in ChIP signal for EZH2, SUZ12 and H3K27me3 peaks, after knockdown of PHF19L (*Figure 3C*). Interestingly, almost no peaks had the opposite trend. This observation was clearly evident for PHF19L target genes, which exhibited a strong increase in PRC2 and H3K27me3 levels after PHF19L depletion (*Figure 3D,E*, and *Figure 3—figure supplement 1C*). We further validated these results by performing ChIP-qPCR in a subset of PHF19L targets, where we confirmed the presence of EZH2 and H3K27me3 in control conditions (shCTR), and the gain of signal in two different PHF19L knockdowns (shPHF19L#4 and shPHF19L#B) (*Figure 3—figure supplement 1D*).

## MTF2 is enriched in chromatin after loss of PHF19L

Two PRC2 subcomplexes (termed PRC2.1 and PRC2.2) have recently been identified; these share the core canonical subunits but are associated with different accessory proteins that can modulate their activity and recruitment to chromatin (*Figure 4A*; *Hauri et al., 2016*). To address the mechanism by which PRC2 occupancy increases in the absence of PHF19L, we investigated other PRC2-

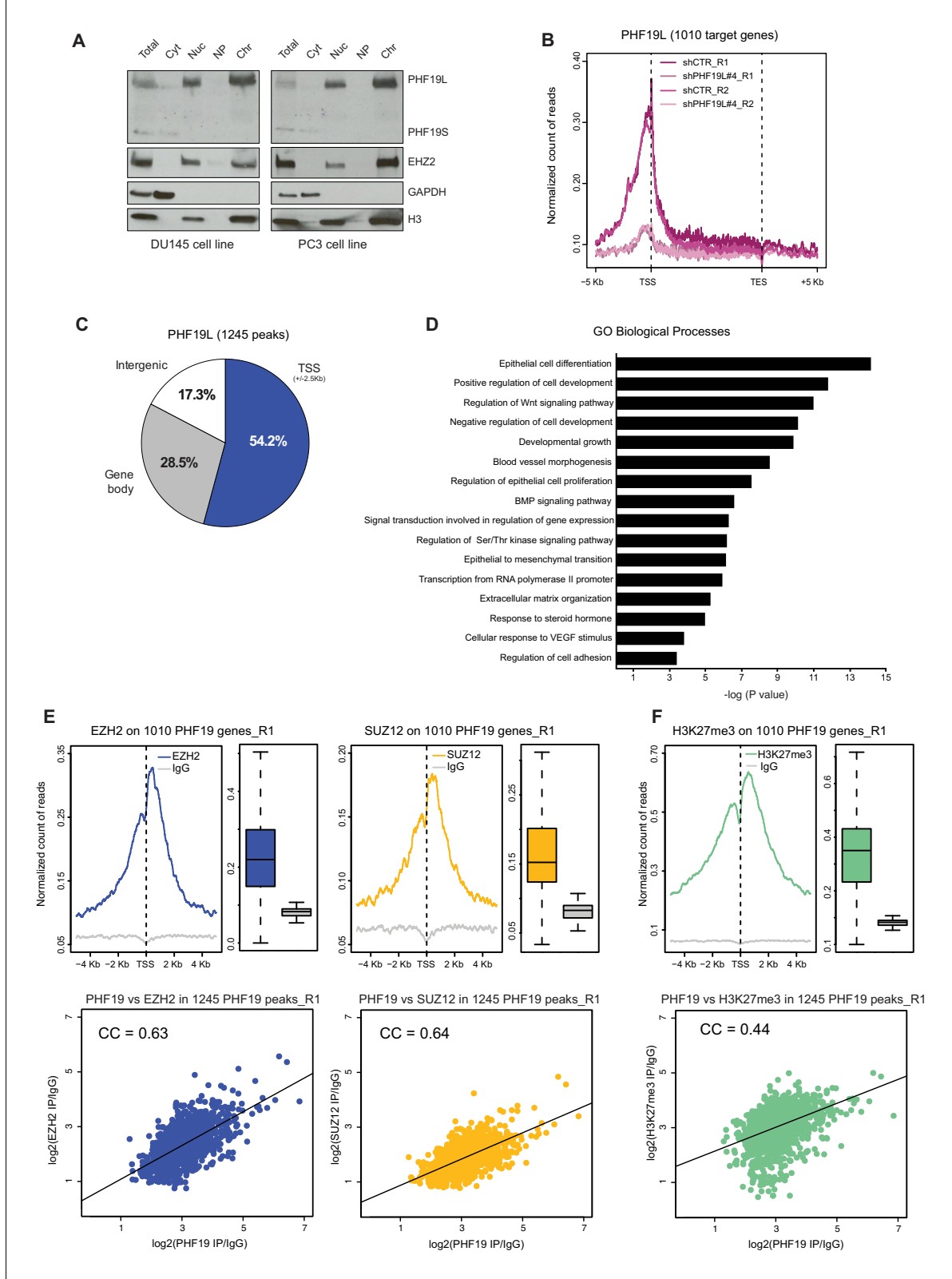

**Figure 2.** PHF19L co-localizes with PRC2 complex on chromatin. (**A**) Cell fractionation followed by Western blot analysis in DU145 (left) and PC3 (right) cells, showing that PHF19L is mainly present in the chromatin and PHF19S, in the cytoplasm. Total, total cell extract; Cyt, cytoplasm; Nuc, nucleus; NP, nucleoplasm; Chr, chromatin. (**B**) Metagene profile showing enrichment of PHF19 ChIP-signal along the 1010 PHF19L target genes in control (shCTR) and PHF19L knockdown (shPHF19L#4) cells, for two independent biological replicates (R1 and R2). Enrichment levels are normalized for the total

*Figure 2 continued on next page*

Figure 2 continued

number of reads of each sample. (C) Pie chart representing the distribution of PHF19L binding sites (ChIP-seq peaks) in the indicated genomic regions. (D) Gene ontology (GO) analysis of biological processes of PHF19L target genes in DU145 cells. (E,F) (Top) TSS (± 5 kb) enrichment plots of the indicated ChIP-seq experiments (Replicate 1) at the 1010 PHF19L target genes in DU145 cells. Boxplots showing the corresponding distribution of values are presented next to each TSS plot. Enrichment levels are normalized for the total number of reads of each sample. (Bottom) Scatter plots comparing the ChIP-seq enrichment signals (IP/IgG) of PHF19L against EZH2 (E, left panel), SUZ12 (E, right panel), or H3K27me3 (F) in the 1245 PHF19L ChIP-peaks. CC is the correlation between each pair of variables (P value $\leq 10^{-16}$ in all 3 cases). P values were computed using Pearson's product-moment correlation.

The online version of this article includes the following figure supplement(s) for figure 2:

**Figure supplement 1.** PHF19L co-localizes with PRC2 complex in chromatin.

associated factors that could potentially compensate for the loss of PHF19L and mediate an accentuated PRC2 recruitment. Gene expression analysis of different PRC2 accessory proteins revealed a specific and significant upregulation of MTF2 (PCL2) expression after depletion of PHF19L (*Figure 4B*). This increase was also verified at the protein level, in both whole cell extracts and the chromatin fraction (*Figure 4C*).

To study whether PHF19L affects MTF2 occupancy in chromatin, we performed ChIP-seq of MTF2 in two biological replicates of control and PHF19L-depleted DU145 cells. Genome-wide analysis of MTF2 occupancy showed a clear enrichment following knockdown of PHF19L, with MTF2 target genes rising from 2011 to 2811 (*Figure 4D*, *Supplementary file 1*). DiffBind analysis between both conditions showed a significant increase in the ChIP-signal for most of the MTF2 peaks in the absence of PHF19L (*Figure 4—figure supplement 1A*). Regarding the PHF19L targets, most of them (73%) coincided with MTF2 targets in control conditions, and this number is further increased (87%) in knockdown cells (*Figure 4D*). PHF19L binding sites exhibited a strong increase in MTF2 signal after PHF19L loss (*Figure 4E,F*). Furthermore, a general gain in MTF2 levels was also observed in the MTF2 targets that were not occupied by PHF19L (*Figure 4—figure supplement 1B*). We validated these results by ChIP-qPCR in a subset of PHF19L target genes using two different shRNAs (*Figure 4—figure supplement 1C*). Importantly, this effect was specific for MTF2 and not a general mechanism affecting all PRC2 associated factors, as the occupancy of JARID2 did not change in the absence of PHF19L (*Figure 4—figure supplement 1D*).

To further confirm the interplay between the occupancy of PHF19L and MTF2, we ectopically expressed a FLAG-tagged PHF19L in cells depleted of PHF19L (shPHF19L#B, which targets the 3'UTR of endogenous PHF19L) as well as in control DU145 cells (shCTR). ChIP analysis indicated that PHF19L binding was diminished in knockdown cells but was rescued upon PHF19L re-expression. Concomitantly, the opposite pattern was observed for MTF2 recruitment, with an increase in its levels in the absence of PHF19L, and a decrease to basal levels (or even lower) after PHF19L overexpression (*Figure 4G*).

Together, these results indicate that, in prostate cancer, PHF19L restricts an excessive occupancy of MTF2 at chromatin, suggesting that MTF2 could be responsible for the increase in PRC2 recruitment and activity following PHF19L depletion.

## PHF19L regulates the expression of genes essential for tumor growth, invasiveness, and metastasis

To understand the functional role of PHF19L in prostate cancer, we carried out global transcriptome analysis by RNA-seq in cells depleted of PHF19L (shPHF19L#4) or PHF19S (shPHF19S#168) and in DU145 control cells (shCTR). Loss of PHF19L resulted in upregulation of 652 genes and downregulation of 847 genes (*Figure 5A*, *Supplementary file 2*). Conversely, depletion of PHF19S had no significant impact on gene expression, with an almost unaltered transcriptome profile (*Figure 5—figure supplement 1A*). Gene Ontology (GO) analysis showed that genes upregulated after PHF19L loss were mainly involved in signaling pathways such as Ser/Thr kinases and ERBB, response to hypoxia, cell migration, extracellular matrix organization, and angiogenesis (*Figure 5B*, upper panel). GO annotation of genes downregulated after PHF19L loss indicated enrichment in those involved in the interferon pathway, development, morphogenesis, and signaling pathways (*Figure 5B*, lower panel). A closer analysis of the transcriptional changes revealed deregulation of essential genes involved in control of cell proliferation, several members of dual-specificity phosphatases (DUSPs)

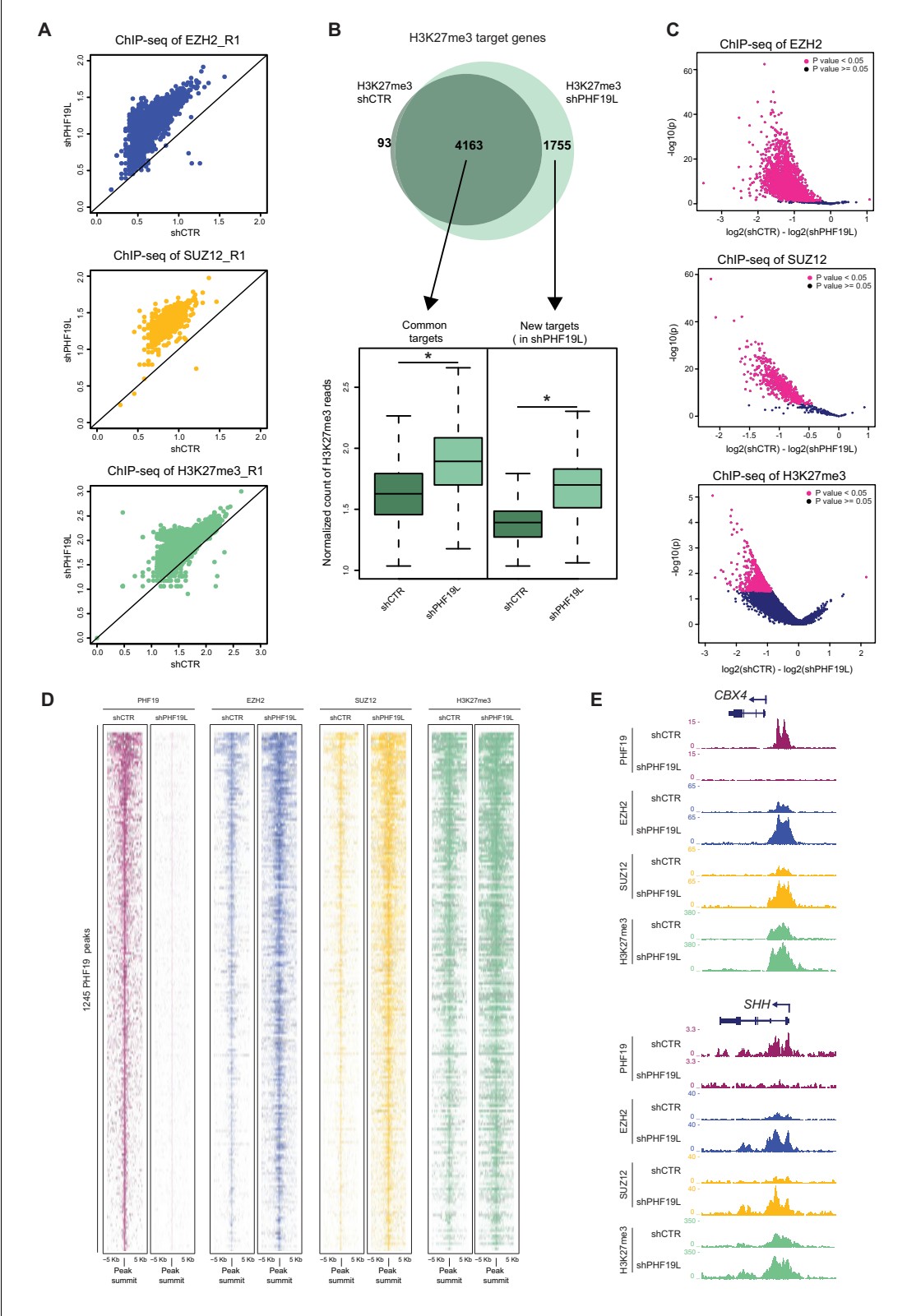

**Figure 3.** Increase of PRC2 occupancy upon PHF19L depletion. (**A**) Scatter plots showing correlation of EZH2, SUZ12, and H3K27me3 ChIP-seq reads in shCTR versus shPHF19L#4 DU145 cells. Each dot in the plot corresponds to the number of ChIP-seq reads normalized by the number of fly spike-in reads of each sample for each target gene. Data correspond to replicate 1 (R1). Upper panel: EZH2 target genes; middle panel: SUZ12 target genes; lower panel: H3K27me3 target genes. After depletion of PHF19L a significant increase in EZH2, SUZ12 and H3K27me3 signal was observed. P

*Figure 3 continued on next page*

*Figure 3 continued*

value $\leq 10^{-16}$ in all cases. P values were computed using Wilcoxon test (two-sided). (B) (Top) Venn diagram showing the overlapping of H3K27me3 target genes in control (shCTR) and PHF19L-depleted (shPHF19L#4) DU145 cells (P value $\leq 10^{-16}$, Fisher's exact test). (Bottom) Boxplot showing H3K27me3 ChIP-seq signal intensity for common genes and new targets in control (shCTR) and PHF19L knockdown (shPHF19L#4) DU145 cells. The increase of signal is significant in all cases (P value $\leq 10^{-16}$, Wilcoxon test, two-sided). Values associated to the peaks were normalized by the total number of fly spike-in reads of each ChIP-seq experiment. (C) Volcano plots of the EZH2 (top), SUZ12 (middle), and H3K27me3 (bottom) ChIP-seq peaks, showing significant changes in signal upon PHF19 loss, as reported by DiffBind using two biological replicates for each condition (shCTR and shPHF19L#4) (P value < 0.05). For each plot, the union of all peaks of the corresponding ChIP (peaks called in shCTR and peaks called in shPHF19L) was considered. The *x*-axis represents the difference in the number of reads between the shCTR and the shPHF19L#4, considering both replicates. The *y*-axis represents the significance of the peaks (-log P value). (D) ChIP-seq heatmap showing the distribution of the PHF19L, EZH2, SUZ12, and H3K27me3 reads on PHF19L peaks (peak summit ± 5 kb) in control and PHF19L-depleted (shPHF19L#4) DU145 cells. Enrichment levels are normalized for the total number of spike-in reads of each sample. Peaks are ranked by the intensity of PHF19 signal in the control condition. (E) UCSC genome browser screenshot of PHF19, EZH2, SUZ12, and H3K27me3 ChIP-seq-profiles from control or PHF19L-depleted (shPHF19L#4) DU145 cells in two representative PHF19L target genes.

The online version of this article includes the following figure supplement(s) for figure 3:

**Figure supplement 1.** Increase of PRC2 occupancy after PHF19L depletion.

that negatively regulate MAP kinases and cell growth (*Caunt and Keyse, 2013*; *Zhai et al., 2014*; *Arnoldussen and Saatcioglu, 2009*), and a set of genes that are usually upregulated upon hypoxia but known to drive angiogenesis and metastasis such as VEGFA, VEGFC (*Sullivan and Graham, 2007*; *Vergis et al., 2008*), CXCR4 (*Darash-Yahana et al., 2004*), and LOX (*Erler et al., 2006*). We validated the changes in the expression of several of these genes by RT-qPCR in both DU145 cells (*Figure 5C*) and PC3 cells (*Figure 5—figure supplement 1B*). To examine whether these effects were directly related to loss of PHF19L, we performed rescue experiments by stably overexpressing PHF19L in knockdown cells (shPHF19L#B, which targets the 3' UTR of endogenous PHF19L), and evaluated the expression of a panel of up- or downregulated genes by RT-qPCR. Changes in gene expression observed in PHF19L- depleted cells were reversed after overexpression of exogenous PHF19L (*Figure 5D*).

Comparative analysis between differential expression data and ChIP-seq results revealed that only 9% of the PHF19L direct targets (91/1010) were deregulated, with 59 genes upregulated and 32 downregulated (*Figure 5E*), suggesting that indirect or secondary effects play an essential role in controlling changes in gene expression after PHF19L depletion. Nevertheless, among the PHF19L targets, we found several downregulated genes directly implicated in inhibition of tumor progression and metastasis in prostate cancer, such as *IGFBP3* (*Mehta et al., 2011*) and *NDRG2* (*Gao et al., 2011*; *Figure 5E*; *Figure 5—figure supplement 1C* left panel). On the other hand, the upregulated PHF19L direct targets included several genes typically induced by hypoxia that are required for homing and establishment of pre-metastatic niche, such as *LOX* (*Erler et al., 2006*), *CXCR4* (*Darash-Yahana et al., 2004*), *EGLN3* (*Henze et al., 2010*), and *ADM* (*Zhang et al., 2017*). We also observed upregulation of Hedgehog signaling pathway components (e.g. *BMP4*, *SHH*, and *WNT7A*) as well as of genes that drive the epithelial-to-mesenchymal transition (*HEY1*, *HMGA2*, and *SOX9*) (*Francis et al., 2018*; *Shi et al., 2016*; *Chen et al., 2010*; *Figure 5E*). Moreover, as the ChIP-seq results revealed that about 17% of PHF19L binding sites are located in intergenic regions, we cannot exclude the possibility that PHF19L binds also to enhancer elements, thus regulating the expression of neighboring genes. Indeed, we observed substantial enrichment in H3K4me1, a histone mark associated to enhancers, in these intergenic peaks (*Figure 5—figure supplement 1D*). Interestingly, we found several examples of intergenic PHF19L binding sites located in the vicinity of downregulated genes (*Figure 5—figure supplement 1C* right panel).

## Depletion of PHF19L switches the cells to a less proliferative but more aggressive phenotype

Global gene expression analysis after depletion of PHF19L showed deregulation of multiple genes involved in control of cell cycle and growth (*Figure 5C*). Indeed, growth curves of DU145 and PC3 cells revealed that PHF19L loss dramatically decreased cell growth (*Figure 6A,B* and *Figure 6—figure supplement 1A*). BrdU incorporation assays also showed a significant reduction in cell proliferation in knockdown cells as compared with control cells (*Figure 6C* and *Figure 6—figure*

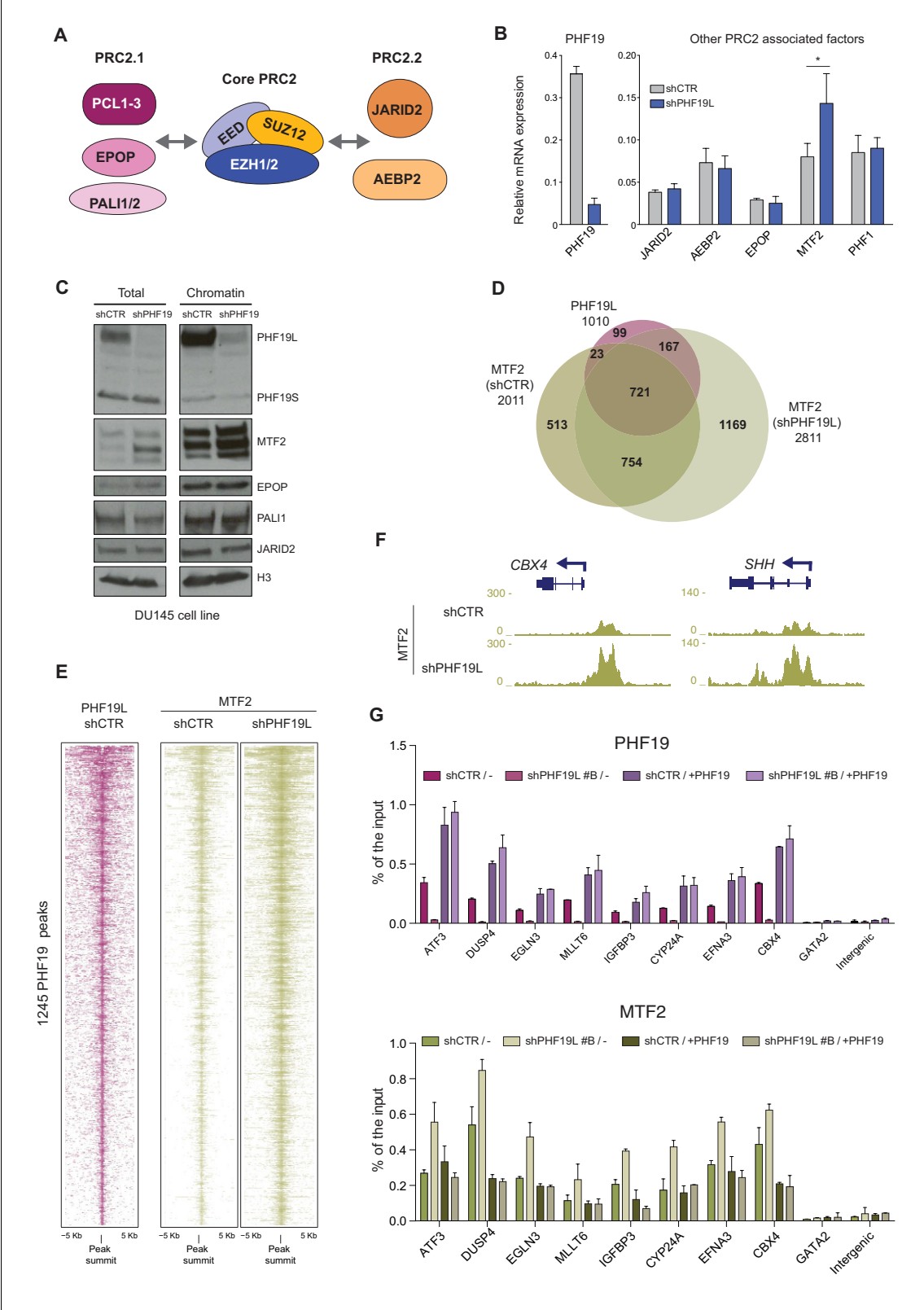

**Figure 4.** MTF2 is enriched in chromatin in the absence of PHF19L. (**A**) Schematic representation of PRC2.1 and PRC2.2 complexes. (**B**) RT-qPCR of PRC2-associated factors in control and PHF19L-depleted (shPHF19L#4) DU145 cells. Expression was normalized to that of the housekeeping gene *RPLPO*. Data are presented as the mean ± SD of three biological replicates. Significance was analyzed through Student's *t*-test. P value was < 0.05 (*). (**C**) Cell fractionation showing specific increase of MTF2 protein in whole cell extracts (total) and in the chromatin compartment after knockdown of

*Figure 4 continued on next page*

*Figure 4 continued*

PHF19L (shPHF19L#4) in DU145 cells. (D) Venn diagram showing overlapping of 1010 PHF19L targets with MTF2 target genes in control (shCTR) and PHF19L-depleted (shPHF19L#4) DU145 cells. (P value $\leq 10^{-16}$; Fisher's exact test). (E) ChIP-seq heatmap of MTF2 on PHF19L peaks (peak summit ± 5 kb) in control and PHF19L-depleted (shPHF19L#4) DU145 cells. Enrichment levels are normalized for the total number of spike-in reads of each sample. Peaks are ranked by the intensity of PHF19 signal in the control condition. (F) UCSC genome browser screenshot showing two examples of genes (*CBX4* and *SHH*) that gain MTF2 after PHF19L knockdown in DU145 cells. (G) Knockdown and rescue of PHF19: ChIP-qPCR experiments of PHF19L and MTF2 in DU145 cells control (shCTR) and PHF19L knockdown (shPHF19L#B, targeting the 3' UTR region), transduced with either FLAG-tagged PHF19L (+PHF19) or FLAG-Empty vector (-). Amplification of the GATA2 and an intergenic region were used as a negative control. Data represent the mean ± SD from two biological replicates.

The online version of this article includes the following figure supplement(s) for figure 4:

**Figure supplement 1.** MTF2 is enriched in chromatin in the absence of PHF19L.

*supplement 1B*). On the other hand, and in concordance with the gene expression data, depletion of PHF19S did not have any effect on cell proliferation (*Figure 6—figure supplement 1C*).

Although PHF19L knockdown cells had reduced proliferation, they had multiple genes upregulated that are associated with invasion and metastasis, as well as several angiogenic factors (*Figure 5B,C*). To explore the potential role of PHF19L in stimulating prostate cancer cell invasion, we performed in vitro invasion assays using matrigel-coated transwells. DU145 cells lacking PHF19L showed a significantly increased capacity to invade the matrigel as compared to wild-type cells (*Figure 6D*). A similar effect was observed for PC3 cells (*Figure 6—figure supplement 1D*).

We then examined the potential implication of PHF19L on angiogenesis. For this, we performed an in vitro endothelial tube formation assay using human umbilical vein endothelial cells (HUVECs) and conditioned medium from control or PHF19L-knockdown cells (*Figure 6E,F*). Our results showed that the number of tubes and nodes significantly increased when HUVECs were incubated in the presence of conditioned medium from knockdown cells, indicating that PHF19L depletion promoted angiogenesis. Taken together, our results indicated that while PHF19L is required to maintain high proliferation rates in prostate cancer cells, its depletion switches cells to a less proliferative but more aggressive phenotype, thereby promoting cell invasion and angiogenesis.

## Discussion

Prostate cancer is one of the leading causes of cancer death in men, and the high mortality rate is mainly due to the development of metastasis. The main therapeutic strategy is based on androgen deprivation (*Jain and Di Croce, 2016*). However, the disease usually progress to metastatic castration-resistant prostate cancer, a hormone insensitive form with very low survival rate (*Mansinho et al., 2018*). A better understanding of the mechanisms that drive the progression to a metastatic disease is a major challenge in the field and could be critical to developing effective therapies. Here, we provide evidence that PHF19L plays an important role controlling prostate cancer progression, and that its depletion results in transcriptional deregulation of multiple genes that are critical for proliferation and metastasis.

Our data show that only the long isoform of PHF19 interacts with PRC2 in prostate cancer cells. PHF19S contains the Tudor domain that is implicated in binding to H3K36me3, but it lacks the predicted nuclear localization signals (*Wang et al., 2004*) and the C-terminal region reported to bind to the PRC2 complex (*Ballaré et al., 2012*). This can explain both the absence of PHF19S at chromatin and its inability to interact with PRC2. We further report for the first time the genome-wide PHF19 target genes in prostate cancer and show its co-localization with PRC2 and H3K27me3. Additionally, we report that loss of PHF19L triggers a global increase in PRC2 occupancy accompanied by a gain in H3K27me3 deposition, suggesting a different function with respect to what has been reported in ESCs (*Ballaré et al., 2012*; *Brien et al., 2012* ). Although the mechanisms of PRC2 recruitment to specific loci are still a matter of intense study, multiple subunits have been identified that associate to PRC2 and regulate its function (*van Mierlo et al., 2019*). Here, we observed an increase in the MTF2 expression and its binding to chromatin after loss of PHF19L. Both MTF2 and PHF19 can mediate PRC2 recruitment (*Li et al., 2017*; *Perino et al., 2018*), but they bind to PRC2 in a mutually exclusive manner (*Hauri et al., 2016*). Therefore, it is conceivable that they compete for PRC2 binding. Loss of PHF19L could activate putative compensatory mechanisms that lead to increasing MTF2

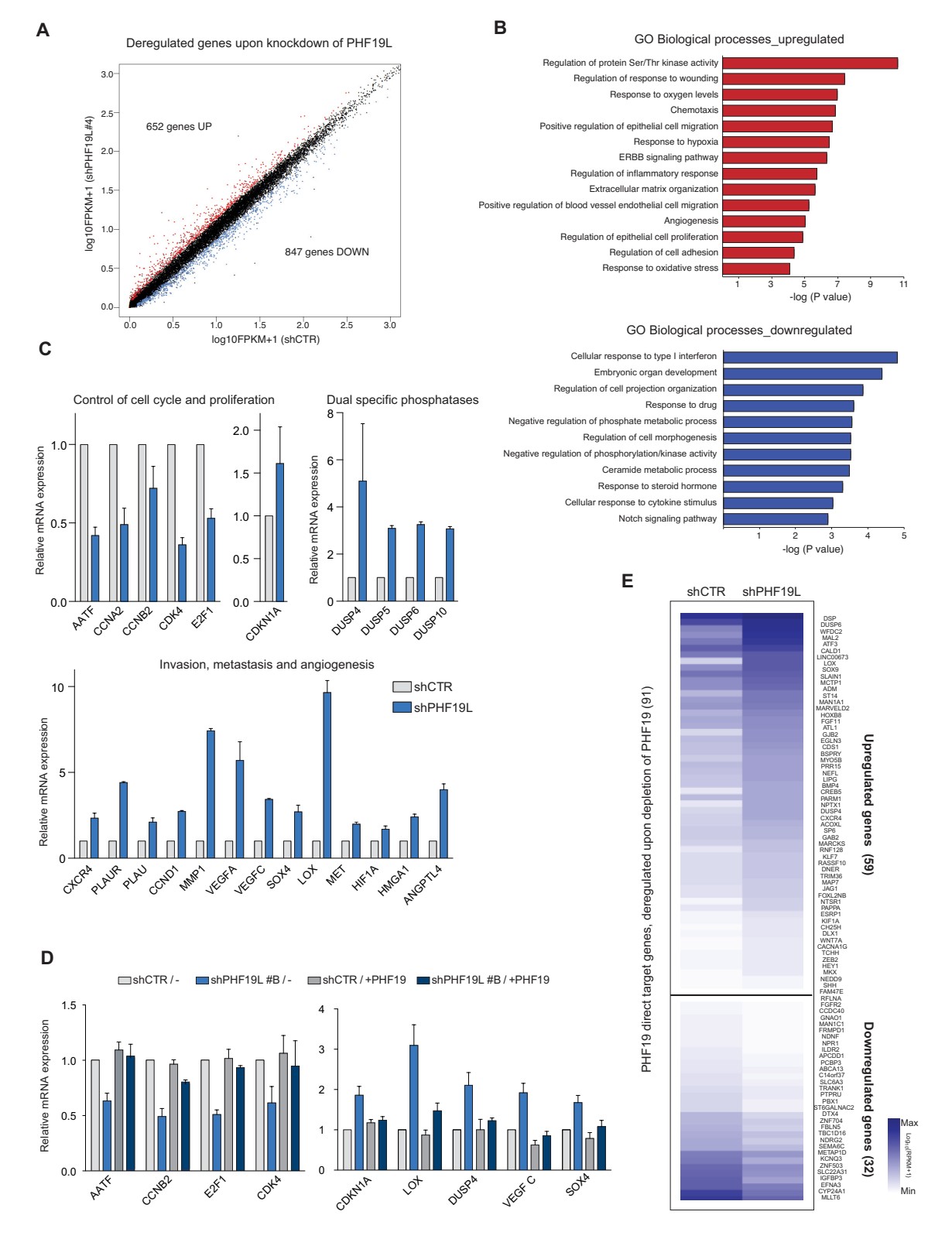

**Figure 5.** PHF19L regulates the expression of genes essential for tumor growth, invasiveness, and metastasis in prostate cancer cells. (**A**) Scatter plot showing changes in gene expression as detected by RNA-seq in PHF19L knockdown (shPHF19L#4) as compared to control (shCTR) DU145 cells. Up- and downregulated genes are highlighted in red and blue, respectively. The remaining genes are shown in black. (**B**) GO analysis of biological processes of upregulated (upper panel) and downregulated (lower panel) genes in DU145 cells after PHF19L knockdown. (**C**) Expression levels of

*Figure 5 continued on next page*

*Figure 5 continued*

selected genes were determined by RT-qPCR in control and PHF19L-depleted (shPHF19L#4) DU145 cells. Results are shown relative to shCTR and are normalized to the housekeeping gene *RPLPO*. Data are presented as mean ± SD of three biological replicates. (D) Expression levels of selected genes were determined by RT-qPCR in DU145 cells control (shCTR) and PHF19L knockdown (shPHF19L#B, targeting the 3' UTR region), transduced with either FLAG-tagged PHF19 (+PHF19) or FLAG-Empty vector. Results are shown relative to shCTR /-and are normalized to the housekeeping gene *RPLPO*. Data are presented as the mean ± SD of two biological replicates. (E) Heatmaps showing RNA-seq signal on PHF19L direct target genes that are transcriptionally up- or downregulated after loss of PHF19L (shPHF19L#4) in DU145 cells.

The online version of this article includes the following figure supplement(s) for figure 5:

**Figure supplement 1.** PHF19L regulates the expression of genes essential for tumor growth, invasiveness, and metastasis in prostate cancer cells.

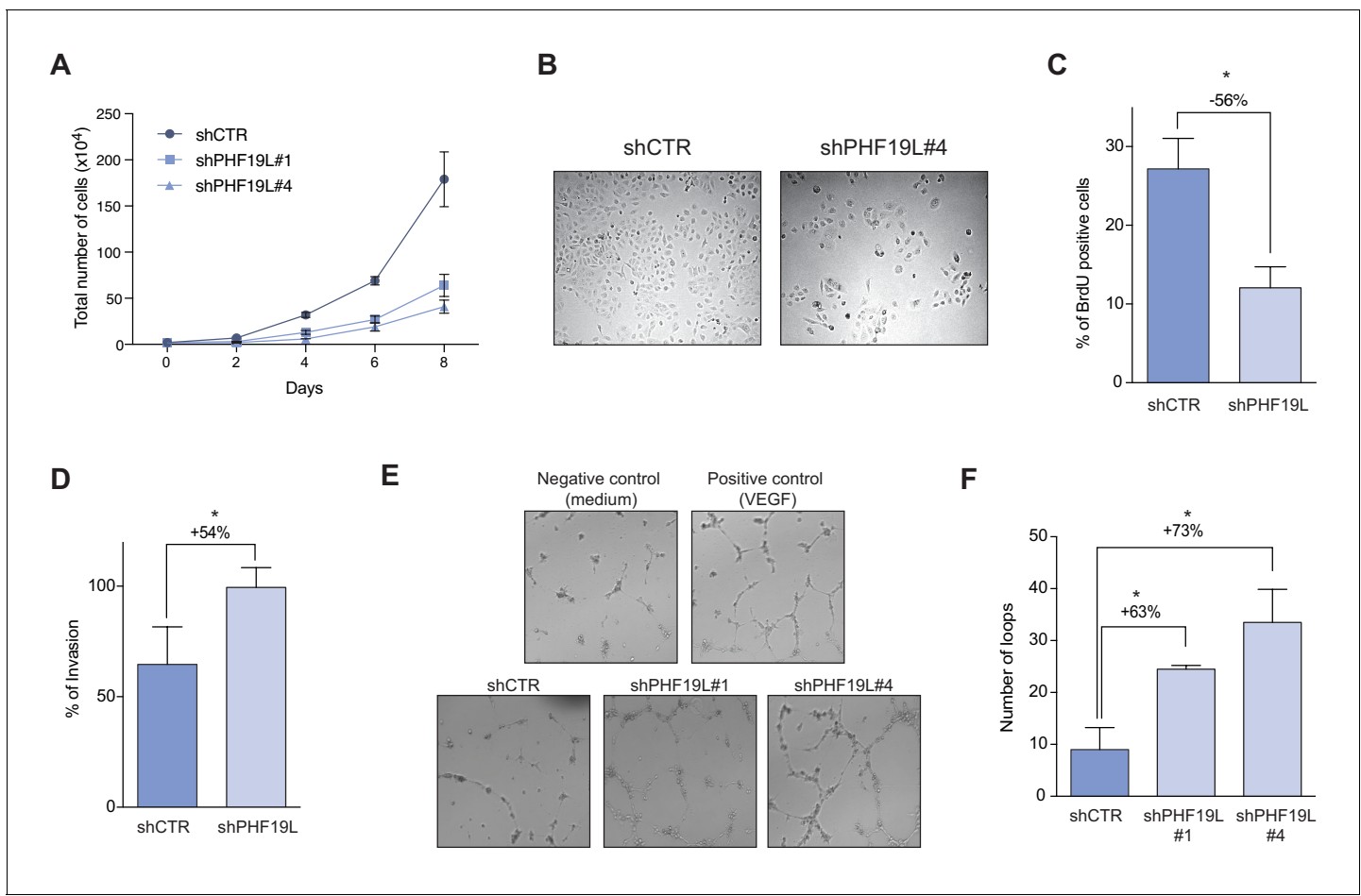

**Figure 6.** Depletion of PHF19L switches the cells to a less proliferative but more aggressive phenotype. (A) Growth curve comparing cell growth of control and PHF19L knockdown (shPHF19L#1 or shPHF19L#4) DU145 cells. Data are presented as mean ± SD of three biological replicates. (B) Phase contrast microscope images of DU145 cells in culture showing decrease in cell growth after PHF19L depletion. (C) Effect of PHF19L depletion (shPHF19L#4) on BrdU incorporation in DU145 cells. Data are presented as mean ± SD of three biological replicates. Significance was analyzed using Student's *t*-test. P value was ≤ 0.05 (*). (D) Transwell invasion assay in control and PHF19L-depleted (shPHF19L#4) DU145 cells. Graph shows the percentage of invasion (mean ± SD) from two biological replicates. Significance was analyzed through Student's *t* test. P value ≤ 0.05 (*). (E,F) Tube formation assays were performed by treating HUVECs with conditioned media from control (shCTR) or PHF19L-depleted (shPHF19L#1 or #4) DU145 cells. (E) Representative images showing increased tube formation in the PHF19L-knockdown as compared to control DU145 cells. VEGF (50 ng/ml) and unconditioned medium were used as positive and negative controls, respectively. (F) Quantification of the number of loops in each condition. Bars represent the mean ± SD from two biological replicates. Significance was analyzed using Student's *t*-test. P value was ≤ 0.05 (*).

The online version of this article includes the following figure supplement(s) for figure 6:

**Figure supplement 1.** PHF19L depletion switches prostate cancer cells to a less proliferative but more aggressive phenotype.

levels. MTF2 likely replaces PHF19L in the PRC2 complex, triggering more efficient PRC2 recruitment and/or increasing its residence time. Thus, PHF19L could have a modulatory role, keeping the levels of PRC2 at bay. Changes in the expression of the PCLs in response to different stimuli or during cancer progression could alter the balance between the PRC2 subcomplexes and thus affect PRC2 binding and activity, providing flexibility to the PRC2 function. Moreover, other PRC2-associated factors with unchanged expression levels after PHF19L loss could also undergo specific changes in chromatin occupancy and therefore affect PRC2 binding. Further studies need to be performed to investigate the interplay of the PRC2 associated subunits in the regulation of PRC2 function in prostate cancer.

PHF19 has been previously reported to be required for cell proliferation in different cellular types (*Ghislin et al., 2012*; *Xu et al., 2015*; *Lu et al., 2018*; *Tao et al., 2018*; *Deng et al., 2018*). The ectopic expression of either PHF19 or MTF2 promotes cell proliferation in human dermal fibroblasts by mediating PRC2 recruitment to the *Ink4A* locus leading to its repression (*Brien et al., 2015*). Here, we report that depletion of PHF19L causes upregulation of *CDKN1A*/p21 and downregulation of several genes essential for the control of cell cycle, such as *CCNA2*, *CCNB2*, *CDK4*, *E2F1*, and *AATF*. Consistent with changes in gene expression, we observed strong decreases in cell growth and proliferation after PHF19L depletion in both prostate cancer cell lines.

Our data show that PHF19L inhibits invasion and angiogenesis in prostate cancer cells. Similar observations have been reported in melanoma cells, in which loss of PHF19L causes them to become more invasive and less proliferative (*Ghislin et al., 2012*). However, contradictory findings have also been reported, showing a positive role of PHF19 promoting migration and invasion in hepatocellular carcinoma (*Xu et al., 2015*) and glioblastoma cells (*Deng et al., 2018*). We found that PHF19L depletion induces upregulation of the HIF-1$\alpha$, a master regulator of the cellular response to low oxygen, as well as of multiple genes associated with hypoxia. Hypoxia is a potent stimulus for tumor progression, as it activates survival mechanisms promoting angiogenesis, invasiveness, metastasis, and the epithelial-to-mesenchymal transition (EMT) (*Sullivan and Graham, 2007*; *Deep and Panigrahi, 2015*). Indeed, by knocking down PHF19L, we observed significant induction of multiple genes known to promote changes in cytoskeleton and adhesion, extracellular matrix remodeling, invasion, and metastasis in prostate cancer (including *SOX9*, *SOX4*, *MMP1*, *PLAU*, *EPCAM*, *CXCR4*, *LOX*, and *MET*) (*Darash-Yahana et al., 2004*; *Erler et al., 2006*; *Francis et al., 2018*; *Liu et al., 2017*; *Pulukuri and Rao, 2008*; *Banyard et al., 2014*; *Pennacchietti et al., 2003*). We also saw upregulation of several genes that stimulate formation of new vessels and regulate the processes of intravasation and extravasation, such as *VEFGA*, *VEGFC*, *MMP1*, and *ANGPTL4* (*Sullivan and Graham, 2007*; *Vergis et al., 2008*; *Gupta et al., 2007*; *Zhang et al., 2012*). In line with these findings, prostate cancer cells lacking PHF19L displayed significantly increased invasiveness, and conditioned medium from PHF19L knockdown cells promoted angiogenesis in in vitro endothelial tube formation assays.

Interestingly, PHF19L depletion triggered both up- and downregulation of direct PHF19L target genes. The repressed gene set included mainly genes involved in signaling and tumor progression, such as *IGFBP3*, a metastasis/angiogenesis suppressor in prostate cancer (*Mehta et al., 2011*). After PHF19L depletion, *IGFBP3* gained MTF2, PRC2 and H3K27me3 in its promoter and was substantially downregulated. On the other hand, the upregulated PHF19L direct targets comprised many hypoxia-responsive genes that can promote invasion and metastasis (e.g. *LOX*, *CXCR4*, *EGLN3*, and *ADM*), as well as genes that induce EMT (e.g. *BMP4*, *SHH*, *WNT7A*, and *SOX9*). It is puzzling why these genes are upregulated, as they also gain PRC2 at their promoters. One possibility would be that EZH2 binds and/or methylates a non-histone protein to promote transcriptional activation. EZH2 was originally identified as the catalytic subunit of PRC2, responsible for tri-methylation of H3K27. However, emerging research have shown non-canonical roles of EZH2, such as methylation of different targets, in both PRC2-dependent and -independent manner (*Gan et al., 2018*). For instance, EZH2 can methylate GATA4, inhibiting its transcriptional activity (*He et al., 2012*), STAT3, leading to its transcriptional activation (*Kim et al., 2013*), and JARID2, triggering activation of PRC2's enzymatic activity (*Sanulli et al., 2015*). It can also control adhesion and migration of neutrophils and dendritic cells through methylation of a key regulatory protein, Talin (*Gunawan et al., 2015*). In addition, EZH2 can interact with transcription factors and act as a co-activator, as has been reported for androgen receptor, β-catenin, ERα, and NF-κB (*Xu et al., 2012*; *Gan et al., 2018*; *Shi et al., 2007*; *Lee et al., 2011*). Further mechanistic studies need to be performed to investigate

if putative EZH2 substrates can account for the response of this subset of PHF19L targets, which becomes upregulated despite gaining PRC2. Nevertheless, our findings show that upon loss of PHF19L, most of the deregulated genes are not direct targets of PHF19L, meaning that these changes are very likely due to secondary or indirect effects, possibly through the activation of signaling pathways. Notably, we also found binding of PHF19L and PRC2 in intergenic regions (23% PHF19L ChIP-peaks), coincident with the enhancer mark H3K4me1. The presence of PRC2 and H3K27me3 in poised enhancers has been reported, with PRC2 playing a role maintaining contacts between enhancers and target genes, and acting as a facilitator of poised enhancers activity during stem cell differentiation (*Cruz-Molina et al., 2017*; *Mas et al., 2018*; *Mas and Di Croce, 2016*). Therefore, it is tempting to speculate that PHF19L can also bind to enhancer elements together with PRC2 in prostate cancer cells and thus plays a role in distal gene regulation.

# Materials and methods

**Key resources table**

| Reagent type (species) or resource | Designation | Source or reference | Identifiers | Additional information |
|---|---|---|---|---|
| Cell line (human) | DU145 | ATCC HTB-81 | RRID:CVCL_0105 | |
| Cell line (human) | PC3 | ATCC CRL-1435 | RRID:CVCL_0035 | |
| Cell line (human) | RWPE1 | ATCC CRL-11609 | RRID:CVCL_3791 | |
| Antibody | PHF19 (rabbit, polyclonal) | Cell Signaling | Cell Signaling #77271; RRID:AB_2799892 | WB (1:1000), ChIP (1:50, 5 µl/chip), IP (1:500, 5 µl/IP) |
| Antibody | EZH2 (rabbit, monoclonal) | Cell Signaling | Cell Signaling #5246; RRID:AB_10694683 | WB (1:1000), ChIP (1:50, 5 µl/chip), IP (1:500, 5 µl/IP) |
| Antibody | SUZ12 (rabbit, monoclonal) | Abcam | Abcam #ab12073; RRID:AB_442939 | ChIP (5 µg/chip) |
| Antibody | H3K27me3 (rabbit, polyclonal) | Millipore | Millipore #07–449; RRID:AB_310624 | ChIP (5 µg/chip) |
| Antibody | MTF2 (rabbit, polyclonal) | Proteintech | Proteintech 16208–1-AP; RRID:AB_2147370 | WB (1:1000), ChIP (5 µg/chip) |
| Antibody | JARID2 (rabbit, polyclonal) | Novus | Novus #NB100-2214; RRID:AB_10000529 | WB (1:1000) |
| Antibody | IgG (rabbit, monoclonal) | Abcam | Abcam #ab172730; RRID:AB_2687931 | ChIP (5 µg/chip) |
| Antibody | EPOP (rabbit polyclonal) | ActiveMotif | Active Motif #61753; RRID:AB_2793758 | WB (1:1000) |
| Antibody | PALI1 (rabbit polyclonal) | Generated in Adrian Bracken's laboratory (*Conway et al., 2018*) | | WB (1:500) |
| Antibody | *Drosophila* H2Av | Active Motif | Active Motif #61686; RRID:AB_2737370 | ChIP (1 µg/chip) |

*Continued on next page*

*Continued*

| Reagent type (species) or resource | Designation | Source or reference | Identifiers | Additional information |
|---|---|---|---|---|
| Antibody | GAPDH (mouse monoclonal) | Santa Cruz Biotechnology | Santa Cruz #sc32233; RRID:AB_627679 | WB (1:5000) |
| Antibody | H3 (rabbit polyclonal) | Abcam | Abcam #ab1791 RRID:AB_302613 | WB (1:2000) |
| Recombinant DNA reagent | Plasmid: MISSION pLKO.1-puro Empty Vector Control | Addgene | Addgene: SHC001 | |
| Recombinant DNA reagent | Plasmid: Plko.1-Puro_shPHF19L#1 | This study | | See *Supplementary file 3* |
| Recombinant DNA reagent | Plasmid: Plko.1-Puro_shPHF19L#4 | This study | | See *Supplementary file 3* |
| Recombinant DNA reagent | Plasmid: Plko.1-Puro_shPHF19L#B | This study | | See *Supplementary file 3* |
| Recombinant DNA reagent | Plasmid: Plko.1-Puro_shPHF19L#168 | This study | | See *Supplementary file 3* |
| Recombinant DNA reagent | Plasmid: Plko.1-Puro_shPHF19L#55 | This study | | See *Supplementary file 3* |
| Sequence-based reagent | RT-qPCR primers | This study | | See *Supplementary file 3* |
| Sequence-based reagent | ChIP-qPCR primers | This study | | See *Supplementary file 3* |
| Commercial assay or kit | ChIP-IT High Sensitivity Kit | Active Motif | Active Motif #53040 | |
| Commercial assay or kit | QIAquick PCR purification kit | Qiagen | Qiagen #28106 | |
| Commercial assay or kit | RNeasy Plus Mini Kit | Qiagen | Qiagen #74134 | |
| Commercial assay or kit | APC BrdU Flow Kit | BD Pharmingen | BD #552598 | |
| Software, algorithm | Bowtie | PMID:19261174 | RRID:SCR_005476 | |
| Software, algorithm | MACS | PMID:18798982 | RRID:SCR_013291 | |
| Software, algorithm | DiffBind | PMID:22217937 | RRID:SCR_012918 | |
| Software, algorithm | R software | *R Development Core Team, 2019* | RRID:SCR_001905 | |
| Software, algorithm | UCSC genome browser | PMID:29106570 | RRID:SCR_005780 | |
| Software, algorithm | Enrichr | PMID:27141961 | RRID:SCR_001575 | |
| Software, algorithm | TopHat | PMID:19289445 | RRID:SCR_013035 | |
| Software, algorithm | Cufflinks | PMID:22383036 | RRID:SCR_014597 | |
| Software, algorithm | SeqCode | http://ldicrocelab.crg.eu/ | RRID:SCR_018070 | Applications to generate ChIP-seq meta-plots, heat maps and boxplots of counts |

## Cell lines and cell culture

The PC3 and DU145 prostate cancer cell lines are derived from bone and brain metastasis of prostate adenocarcinoma, respectively. They were cultured in Dulbecco's Modified Eagle Medium (DMEM) supplemented with 10% fetal bovine serum (FBS) (Gibco), 1 × L glutamine (Gibco) and 1 × penicilin/streptomycin (Gibco). RWPE1, a normal prostate epithelial cell line, was cultured in keratinocyte serum free medium (K-SFM) supplemented with 0.05 mg/ml bovine pituitary extract (BPE) and 5 ng/ml human recombinant epidermal growth factor (EGF). Cells used in this study were authenticated cell lines obtained from ATCC. Mycoplasma contamination tests gave negative results on all the cells used.

## Cell growth curve

About 20,000 cells were seeded in 1 ml medium in a 12-well plate for each condition (day 0); medium was changed every two day. Cells were counted on days 2, 4, 6, and 8, under a light microscope using a counting chamber with Trypan blue staining to exclude dead cells.

## BrdU cell proliferation assay

PC3 and DU145 cells were treated with 10 μM of BrdU solution for 30 min and 2 hr, respectively, and then analyzed for BrdU incorporation using APC BrdU Flow Kit (BD Pharmingen) according to manufacturer´s protocol. The percentage of BrdU-positive cells was analyzed by a Becton Dickinson FACSCanto flow cytometer.

## Transswell migration and invasion assay (Boyden chamber assay)

Cell invasion and migration experiments were performed using cell culture inserts (8 μm pore size, Transparent PET membrane; Corning) as the upper chamber, on 12-well Multiwells (Corning). For the invasion assays the upper membranes were coated with 100 μl Matrigel (Corning #356230). A total of $2 \times 10^5$ DU145 cells (shCTR or shPHF19) in 400 μl of serum-free medium were seeded on the upper chamber. The lower chambers were filled with 1.4 ml of growth medium supplemented with 10% FBS. After a 24 hr incubation at 37°C and 5% CO2, non-migrated cells were gently removed from the upper side of the membrane using a cotton swab. The inserts were transfer to 70% ethanol for 10 min to allow cell fixation and air-dried for 15 min. The cells were stained with crystal violet 0.2% for 10 min and then with DAPI (1 μg/ml) for nuclei visualization. The migration assay was performed similarly to the invasion assay but without coating the upper membranes with Matrigel. Images of the stained cells were then captured under bright-field microscopy and cells from at least five randomly selected fields were counted for each experiment. Percentage of invasion was calculated as the ratio of cells that passed through the Matrigel-coated membrane divided by cells that migrated through the uncoated membrane ×100.For PC3 cells, FluoroBlok 24-Mutliwell Insert Plates (8 μm pore size, PET membrane, Corning) coated or not with Matrigel, were used. Cells were pre-labelled with 10 μg/ml DilC$_{12}$ (*Pasini and Di Croce, 2016*) (BD Biosciences) fluorescent dye overnight at 37°C. The assay was performed according to manufacturer´s protocol. Fluorescence was read at wavelength of 549/565 nm (Ex/Em) using Tecan Infinite 200 Pro microplate reader.

Percentage of Invasion was calculated as the ratio of the mean RFU of cells that passed through the Matrigel coated membrane divided by the mean RFU of cells that migrated through uncoated membrane ×100. RFU = relative fluorescence units.

## In vitro HUVEC tube-formation assay

In vitro HUVEC tube formation assay was performed following a previously published protocol (*Jm and Lung, 2012*). Briefly, DU145 cells, shCTR and shPHF19L, were seeded and grown to 40% confluence. The growth medium was replaced with serum free DMEM and cells incubated for 24 hr. The conditioned medium (CM) was then harvested. Umbilical vein endothelial cells (HUVECs), grown at 70–80% confluency, were serum starved in Medium 200PRF for 3 hr prior to performing the tube formation assay. After serum starvation, cells were collected and resuspended in serum free DMEM at $4 \times 10^5$ cells/ml. 500 μl of this HUVEC cell suspension were centrifuged at 4000 rpm for 3 min and resuspended in 500 μl of CM obtained from shCTR or shPHF19L DU145 cells, and supplemented with FBS to a final concentration of 1%. Cell suspension was plated in a 96-well plate (100

µl/well) pre-coated with growth factor-reduced Matrigel and incubated at 37°C, 5% CO2 for 6 hr. The cells were then visualized under the light microscope and images of the capillary network were taken.

## Calcium phosphate transfection

HEK-293T ($2.5 \times 10^6$) or Phoenix-AMPHO ($2 \times 10^6$) cells were plated onto a p10 plate. The following day, the calcium phosphate-DNA precipitates were prepared by pooling together the plasmids in 0.25 M CaCl2. While vortexing, calcium phosphate-DNA solution was added dropwise to an equal volume of HBS 2× (HEPES-buffered saline solution, pH 7.05: 0.28 NaCl, 0.05 M HEPES, and 1.5 mM Na2HPO4) at RT. After 30 min at RT, the solution was added to the HEK-293T cells for lentivirus production or Phoenix-AMPHO cells for retrovirus production. Cells were incubated 16 hr with the transfection mix, after that the medium was replaced. At 48 hr and 72 hr, medium containing viral particles was collected and filtered (0.45 µm filter).

## Lentivirus production and infection

Lentivirus were produced by transfecting HEK-293T cells with 10 µg CMVDR-8.91, 5 µg pCMV-VSV-G and 7 µg pLKO-shRNA (shCTR or shPHF19, Sigma) plasmids, using the calcium phosphate method. Medium containing the lentiviral particles was used to infect target cells 24 hr and 48 hr after seeding. Infected cells were selected with puromycin (1 µg/ml for DU145, and 2 µg/ml for PC3 cells).

## Retrovirus production and infection

For overexpression of FLAG-PHF19L or -PHF19S, retroviral vector pMSCV-puro (or pMSCV-neo for rescue experiments) was used. cDNA from PHF19L and PHF19S were cloned into pCMV-FLAG which was then used to PCR purify the FLAG-PHF19L and FLAG-PHF19S sequences and clone them into pMSCV. The pMSCVpuro-FLAG-PHF19S plasmid was digested with EcoRI to release PHF19S and generate pMSCVpuro-FLAG-empty vector. Retrovirus were produced by transfecting Phoenix cells with 10 µg of pMSCV-puro (or neo)-FLAG (empty/PHF19L/PHF19S) and 5 µg pCMV-VSV-G using the calcium phosphate method. For the infection, 2 ml of the medium containing the viral particles were added to each well in 6-well plate containing target cells, plus Polybrene 5 g/µl. The plates were spun at 1000 × g, 32°C for 90 min. The infected cells were then incubated for 3 hr at 37°C and the medium was replaced. Following a second round of infection, cells were selected with 2 µg/ml puromycin. shRNAs specifically targeting PHF19S were designed using http://hannonlab.cshl.edu/GH_shRNA.html and cloned into XhoI and EcoRI site of pLMP/MLP-Puro-GFP retroviral vector. Retrovirus were produced by transfecting Phoenix cells with 10 µg pMLP plasmid (MLP or shPHF19S) and 6.6 µg pCMV-VSV-G using the calcium phosphate method. For the infection, medium containing the viral particles was used to infect target cells. Three rounds of infections (2 hr each) were performed. Infected cells were selected using puromycin (2 µg/ml).

## Rescue experiments

DU145 cells were stably transduced with pMSCV-neo-FLAG-PHF19L or FLAG-empty by spinoculation as previously described (see Retrovirus production and infection). After neomycin selection, cells were infected with pLKO-shRNAs (shPHF19L#B or shCTR) and infected cells were selected with puromycin (see Lentivirus production and infection). shPHF19L#B targets the 3' UTR of PHF19L and therefore only affects the expression of endogenous PHF19L.

## FLAG affinity purification and mass spectrometry

PC3 cells stably expressing FLAG-tagged constructs (PHF19L, PHF19S, or empty vector), were incubated in lysis buffer (50 mM Tris-HCl pH 7.5, 150 mM NaCl, 1 mM EDTA, 1 mM EGTA, 0.5% Triton X-100, plus protease inhibitors) for 30 min on a rotating wheel at 4°C, followed by sonication for 3 cycles (10' ON/30' OFF) in a Bioruptor (Diagenode). The lysates were then clarified by centrifugation (15,000 × g, 30 min, 4°C). Cell lysates (5 mg) were incubated with 100 µl FLAG M2 affinity gel (SIGMA) 3 hr at 4°C. The beads were then washed three times with lysis buffer and twice with TBS. Two rounds of elution were performed with a buffer containing 6 M urea and 200 mM NaHCO$_3$. Samples were eluted using a Thermoshaker at 1000 rpm, 30 min each, at room temperature. Eluted

complexes were analysed by mass spectrometry (MS) at the UPF/CRG Proteomics Unit. About 10% of the eluates were used to validate the FLAG immunoprecipitation by Western blot. Proteins were considered to be interactors only when two or more peptides were assigned to the protein in two independent replicates, and none of its peptides were found in FLAG-empty.

## Preparation of protein extracts and western blot

Cell extracts for Western blot analysis were prepared in lysis buffer (25 mM Tris–HCl pH7.6, 1% SDS, 1 mM EGTA, 1 mM EDTA), incubated 10 min at 95°C, sonicated for 3 cycles (30' ON/30' OFF) in a Bioruptor (Diagenode) and centrifuged for 30 min at 13000 rpm at 4°C. Protein supernatant was quantified by Bradford assay, diluted in Laemmli buffer, and analyzed by SDS–PAGE. Western blot was performed as was previously described (*Santanach et al., 2017*).

## Co-immunoprecipitation (co-IP)

Cells were lysed in IP buffer (50 mM Tris-HCl, pH 7.5, 150 mM NaCl, 1 mM EDTA, 1 mM EGTA, 5 mM $MgCl_2$, 0.5% Triton X-100, plus with protease and phosphatase inhibitors) 30 min at 4°C, and sonicated (3 cycles, 10' ON/30' OFF) in a Bioruptor. The lysates were clarified by centrifugation (15,000 × g, 30 min, at 4°C) and soluble material was quantified by Bradford. Per IP, 60 µl Protein A or G Sepharose 4FastFlow Beads (GE Healthcare) pre-blocked with BSA (0.5 mg/ml) were conjugated to the specific antibodies (5 µg) for 3 hr in a rotating wheel at 4°C. Lysates (2 mg) were incubated overnight with the antibodies conjugated to the beads. The beads were then washed four times with IP buffer and eluted with Laemmli buffer. Eluates were separated over SDS–PAGE gels for Western Blot analysis.

## Cell fractionation

Cell fractionation was performed following Mendez and Stillman's protocol (*Méndez and Stillman, 2000*) with few modifications. Briefly, 25 × 10^6 cells were resuspended in 500 µl Buffer A (10 mM HEPES pH 7.9, 10 mM KCl, 1.5 mM MgCl2, 0.34 M sucrose, 10% glycerol, 1 mM DTT, 0.1% Triton X-100, plus protease/phosphatase inhibitors) and incubated 5 min. on ice (total fraction). After a low-speed centrifugation (5 min, 1,300 × g, 4°C), the supernatant was recovered and further clarified by a high-speed centrifugation (15 min, 15,000 × g, 4°C) to remove cell debris and insoluble aggregates (cytoplasmic fraction). Nuclear pellets were washed in Buffer A and resuspend in 50 µl of Buffer A (nuclear fraction). Nuclei were lysed by adding 500 µl of Buffer B (3 mM EDTA, 0.2 mM EGTA, 1 mM DTT, protease/phosphatase inhibitors), and incubated 30 min on ice. Chromatin was collected by centrifugation (5 min, 1,700 × g, 4°C), washed once in Buffer B, resuspended in 500 µl Buffer B.SDS (25 mM TrisCl pH 7.5; 1% SDS; 1 mM EDTA) and sonicated 4 cycles (15' ON/45' OFF) in a Bioruptor (Diagenode). Supernatant was recovered and further clarified by high-speed centrifugation (15 min, 15,000 × g, 4°C) (nucleoplasmic fraction). Aliquots of each fraction (10% of the volume) were collected, mixed with an equal volume of Buffer B.SDS 2 × and boiled 10 min. Proteins were quantified in the total fraction. Proportional volumes of each fraction were loaded for Western blot.

## Gene expression analysis

RNA was extracted using RNeasy mini kit (Qiagen) following the manufacturer's instructions. cDNA was synthesized by reverse transcription from 1 µg of RNA using qScript cDNA synthesis kit (Quanta Biosciences). Real-time PCR reactions were performed using SYBR Green I PCR Master Mix (Roche) and the Roche LightCycler 480. Expression values were normalized to the housekeeping gene *RPLP0*. All primers used are listed in *Supplementary file 3*. For RNA-seq, RNA samples (triplicates) were quantified, and the quality evaluated using Bioanalyser. Libraries were prepared at the UPF/CRG Genomics Unit, using 1 ug total RNA and sequenced using the Illumina HiSeq2000 sequencer.

## Chromatin immunoprecipitation

Four 15 cm plates for each cell line were prepared at 70–80% confluency. Cells were crosslinked in 1% formaldehyde in DMEM for 10 min at RT. To stop the fixation, glycine was added to a final concentration of 0.125 M and incubated for 5 min at RT. Cell were then washed twice with ice cold PBS, harvested by gently scrapping on ice, centrifuged at 3,000 × g, 5 min, and cell pellets were frozen

at −80°C until use. Chromatin preparation and ChIP experiments were performed with the ChIP-IT High Sensitivity Kit from Active Motif (#53040) according to the manufacturer's instructions. ChIPs were performed using 5 µg/ChIP of the following antibodies: PHF19 (Cell Signaling #77271), EZH2 (Cell Signaling #5246), SUZ12 Abcam #ab12073), H3K27me3 (Millipore #07–449), MTF2 (Proteintech 16208–1-AP), JARID2 (Novus #NB100-2214), and control IgG (Abcam #ab172730). ChIP experiments of EZH2, SUZ12, H3K27me3 and MTF2 in control and PHF19L knockdown condition were performed with spike-in control. For this, an equal amount of *Drosophila melanogaster* S2 cell chromatin was added to each ChIP reaction (2.5% of the DU145 cell chromatin for H3K27me3 ChIPs, and 0.1% for the rest of the ChIPs), together with 1 µg of an antibody against a *Drosophila* specific histone variant, H2Av (Active Motif, catalog no. 61686).

A sample of 2 µl was used for ChIP-qPCR analysis. Real-time PCR reactions were performed using SYBR Green I PCR Master Mix (Roche) and the Roche LightCycler 480. All primers used are listed in *Supplementary file 3*.

For ChIP-seq experiments, library preparation was performed at the UPF/CRG Genomics Unit. The libraries were sequenced using Illumina HiSeq2000 sequencer.

Antibodies, primers and shRNAs used in this study are listed in *Supplementary file 3*.

## Bioinformatics analysis

### ChIP-seq analysis

ChIP-seq samples containing spike-in were mapped against a synthetic genome constituted by the human and the fruit fly chromosomes (hg19 + dm3) and those without spike-in were mapped against the hg19 human genome assembly, using Bowtie with the option –m 1 to discard those reads that could not be uniquely mapped to just one region (*Langmead et al., 2009*). A second replicate of each sample was sequenced to evaluate the statistical significance of the results. MACS was run individually on each replicate with the default parameters but with the shift-size adjusted to 100 bp to perform the peak calling against the corresponding control sample (*Zhang et al., 2008*). DiffBind was initially run over the peaks reported by MACS for each pair of replicates of the same experiment to generate a consensus set of peaks (*Ross-Innes et al., 2012*). Next, DiffBind was run again over each pair of replicates of the same experiment - samples and inputs - to find the peaks from the consensus set that were significantly enriched in both replicates in comparison to the corresponding controls (categories = DBA_CONDITION, block = DBA_REPLICATE and method = DBA_DESEQ2_BLOCK). In all cases, DiffBind peaks with P value < 0.05 and FDR < 0.2 were selected for further analysis. The genome distribution of each set of peaks was calculated by counting the number of peaks fitted on each class of region according to RefSeq annotations. Promoter is the region between 2.5 Kbp upstream and 2.5 Kbp downstream of the transcription start site (TSS). Genic regions correspond to the rest of the gene (the part that is not classified as promoter) and the rest of the genome is considered to be intergenic. Peaks that overlapped with more than one genomic feature were proportionally counted the same number of times. Each set of target genes was retrieved by matching the ChIP-seq peaks in the region 2.5 Kbp upstream of the TSS until the end of the transcripts as annotated in RefSeq. The significance of the overlap between two gene sets was computed using the GeneOverlap R package (Fisher's exact test). Reports of functional enrichments of GO and other genomic libraries were generated using the EnrichR tool (*Kuleshov et al., 2016*). The UCSC genome browser was used to generate the screenshots of each profile.

Aggregated meta-plots showing the average distribution of ChIP-seq reads of each sample against the control (IgG) around the TSS of each target gene (+/-5 Kbp) were generated by counting the number of reads for each region according to RefSeq and then averaging the values for the total number of mapped reads of each sample and the total number of genes in the particular gene set. To generate the aggregated plots showing the distribution of ChIP-seq reads along the body of a metagene derived from a target gene set, each gene was converted into a uniform region of 100 positions to count the number of reads along this region, calculating the mean at each point of the resulting metagene profile afterwards. This graphical representation was integrated into the neighboring genomic region, calculated as described above. The aggregated plot showing the average distribution of PHF19L and H3K4me1 ChIP-seq reads for the collection of intergenic PHF19L peaks was generated by counting the number of reads around the summit of each peak and normalizing

for the total number of peaks in this set. Boxplots showing the ChIP level distribution for each replicate of a particular ChIP-seq experiment against the control (IgG) on a set of genomic peaks were calculated by determining the maximum value on this region at this sample, which was normalized by the total number of reads. To generate the points in the scatterplots that represent the co-occupancy between PHF19L and EZH2/SUZ12/H3K27me3 at each replicate, the ChIP-seq intensities of each pair of samples shown here were calculated by determining the maximum value of each experiment inside the peaks of the final set of PHF19L peaks.

Aggregated plots of ChIP-seq shCTR and shPHF19L#4 samples containing spike-in were generated by counting the number of reads mapped in human for each gene and then normalizing these values for the total number of reads mapped on the fruit fly spike-in genome and the number of targets of the gene list, as previously described *Orlando et al. (2014)*. Boxplots showing the ChIP level distribution for each replicate of a particular ChIP-seq experiment in shCTR and shPHF19L#4 conditions on a set of genes were calculated by determining the maximum value on the region +/-5 Kbp around the TSS of every gene in both samples. The resulting values of the samples including spike-in were corrected by the number of fly reads mapped of the sequencing experiment. Each point on the scatterplots of ChIP-seq intensities between shCTR and shPHF19L#4 conditions of EZH2/SUZ12/H3K27me3 were calculated by determining the maximum value of the sample inside each peak at each condition. These values were normalized by the corresponding number of fly spike-in reads in the same experiment. DiffBind was run for each pair of shCTR and shPHF19L#4 ChIP-seq replicates of EZH2/SUZ12/H3K27me3/MTF2 to identify the set of peaks that were significantly enriched in one of the conditions against the other (P value < 0.05). The heatmaps displaying the density of ChIPseq reads around the summit of each ChIP-seq peak were generated by counting the number of reads in this region for each individual peak and normalizing this value with the total number of mapped reads of the sample or the spike-in control, if available. Peaks on each ChIP heatmap were ranked by the logarithm of the average number of reads in the same genomic region.

### RNAseq analysis

RNA-seq samples in triplicates were mapped against the hg19 human genome assembly using TopHat (*Trapnell et al., 2009*) with the option –g 1 to discard those reads that could not be uniquely mapped in just one region. Cufflinks and Cuffdiff (*Trapnell et al., 2012*) were run to quantify the expression in RPKMs of each annotated transcript in RefSeq and to identify the list of differentially expressed genes for each case (P value $\leq$ 0.05; FC $\geq$ 1.4). Expression values shown in the boxplots correspond to the average RPKMs across the 3 replicates in each condition.

## Accession numbers

Raw data and processed information of the ChIPseq and RNA-seq experiments generated in this article were deposited in the National Center for Biotechnology Information Gene Expression Omnibus (NCBI GEO) repository under the accession number GSE135623.

## Acknowledgements

We thank all the members of Di Croce laboratory for helpful discussions, and V A Raker for scientific editing.

The work in the Di Croce laboratory is supported by grants from the Spanish Ministry of Science and Innovation (AEI, BFU2016-75008-P), "Fundación Vencer El Cancer" (VEC), the European Regional Development Fund (ERDF), Fundació "La Marató de TV3", and from Secretaria d'Universitats i Recerca del Departament d'Economia I Coneixement de la Generalitat de Catalunya (AGAUR, 2017 SGR). PV was supported by "Fundación Científica de la Asociación Española Contra el Cáncer". We acknowledge support of the Spanish Ministry of Science and Innovation through the Instituto de Salud Carlos III and to the EMBL partnership; Centro de Excelencia Severo Ochoa; CERCA Programme / Generalitat de Catalunya.

## Additional information

### Funding

| Funder | Grant reference number | Author |
| --- | --- | --- |
| Spanish Ministry of Science and Innovation | BFU2016-75008-P | Luciano Di Croce |
| Fundación Vencer El Cancer | | Luciano Di Croce |
| European Regional Development Fund | | Luciano Di Croce |
| Fundación Científica Asociación Española Contra el Cáncer | | Pedro Vizan |
| Fundació la Marató de TV3 | | Luciano Di Croce |
| Agència de Gestió d'Ajuts Universitaris i de Recerca | | Luciano Di Croce |
| Generalitat de Catalunya | Secretaria d'Universitats i Recerca del Departament d'Economia I Coneixement | Luciano Di Croce |
| Instituto de Salud Carlos III | | Luciano Di Croce |
| European Molecular Biology Laboratory | | Luciano Di Croce |
| Centro de Excelencia Severo Ochoa | | Luciano Di Croce |
| Generalitat de Catalunya | Centres de Recerca de Catalunya | Luciano Di Croce |

The funders had no role in study design, data collection and interpretation, or the decision to submit the work for publication.

### Author contributions

Payal Jain, Cecilia Ballare, Conceptualization, Data curation, Formal analysis; Enrique Blanco, Formal analysis; Pedro Vizan, Data curation; Luciano Di Croce, Conceptualization, Supervision, Funding acquisition, Project administration

### Author ORCIDs

Luciano Di Croce (iD) https://orcid.org/0000-0003-3488-6228

### Decision letter and Author response

Decision letter https://doi.org/10.7554/eLife.51373.sa1
Author response https://doi.org/10.7554/eLife.51373.sa2

## Additional files

### Supplementary files

• Supplementary file 1. Summary table of ChIP-seq data for all ChIPs performed in this study: ChIP-seq peaks and target genes.

• Supplementary file 2. Summary table of RNAseq data: Lists of upregulated and downregulated genes detected in PHF19L knockdown as compared to control DU145 cells.

• Supplementary file 3. Lists of primers for RT-qPCR and ChIP-qPCR, shRNA sequences and antibodies used in this study.

• Transparent reporting form

## Data availability

Raw data and processed information of the ChIPseq and RNA-seq experiments generated in this article were deposited in the National Center for Biotechnology Information Gene Expression Omnibus (NCBI GEO) repository under the accession number GSE135623.

The following dataset was generated:

| Author(s) | Year | Dataset title | Dataset URL | Database and Identifier |
|---|---|---|---|---|
| Blanco E, Ballare C | 2020 | PHF19 mediated regulation of proliferation and invasiveness in prostate cancer cells | http://www.ncbi.nlm.nih.gov/geo/query/acc.cgi?acc=GSE135623 | NCBI Gene Expression Omnibus, GSE135623 |

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
