## [Decision Letter]

**Acceptance summary:**

The PRC2 complex – responsible for the deposition and maintenance of the repressive H3K27me3 chromatin mark – exists in different configurations, which can be altered in cancer. The authors report here that one PRCS2 co-factor, PHF9, is overexpressed in cellular models of prostate cancer. Surprisingly, upon knockdown of PHF19, PRC2 binding and deposition of H3K27me3 are enhanced, through the compensatory action of MTF2. This phenomenon seems important for proliferation and invasiveness of prostate cancer cells. In aggregate, this study reveals that the balance between PHF19 and MTF2 modulates PRC2 binding and activity on chromatin state and gene expression, and this could influence tumorigenesis.

**Decision letter after peer review:**

Thank you for submitting your article "PHF19 mediated regulation of proliferation and invasiveness in prostate cancer cells" for consideration by *eLife*. Your article has been reviewed by Kevin Struhl as the Senior Editor, a Reviewing Editor, and three reviewers. The following individual involved in review of your submission has agreed to reveal their identity: Celine Vallot (Reviewer #2).

The reviewers have discussed the reviews with one another and the Reviewing Editor has drafted this decision to help you prepare a revised submission.

Summary:

The PRC2 complex – responsible for the deposition and maintenance of the repressive H3K27me3 chromatin mark – exists in different configurations, which can be altered in cancer. The authors report here that one co-factor of PRC2, PHF9, is overexpressed in cellular models of prostate cancer but only the long isoform of PHF19 (PHF19L) associates with PRC2. Surprisingly, knockdown of PHF19 enhances PRC2 binding and deposition of H3K27me3, which the authors link to compensation by MTF2. This phenomenon seems important for proliferation and invasiveness of prostate cancer cells. Studying the role of PHF19 in cancer cells is of importance to understand processes of gene misregulation in tumorigenesis. PHF19 had been previously studied in a variety of cancer models but this is the first report in prostate cancer. Most importantly, this study reveals that the balance between PHF19 and MTF2 modulates PRC2 binding and activity on chromatin and gene expression, and this could influence tumorigenesis. However, some conclusions are not sufficiently supported by experiments and statistical tests are too often lacking. Please address the following concerns in a revised version of the manuscript:

Essential revisions:

1) Two important and unanimous critics relate to the overall lack of information about replicates and insufficient statistical assessments.

Information on replication should be more comprehensive and systematically provided in Figure legends. For example (and not the least important), it is not known as to whether ChIP-seq experiments were performed in replicates or not. The use of spike-in is appreciated, but if there is no biological replicate, this should be acknowledged by the authors as a clear disclaimer in the text. This is particularly important as the conclusion from the ChIP-seq is at the center of the main conclusion of the study and also, because the increase in H3K27me3 (Figure 3A, right) is minor and would only be convincing if observed across replicates. On this point, the authors state that the increase is small albeit "significant", but no statistical test is presented.

Indeed, there is generally poor statistical back-up of the results. Some p-values are provided for Figure 4B and Figure 6C, but most of the other comparisons lack statistical testing. For example, in Figure 4D, no statistical testing is done to evaluate the significance of the overlaps. Please provide appropriate statistical tests. Moreover, it is suggested that all p-values should be corrected for multiple testing – Benjamini-Hochberg procedure for example.

2) About the conclusion that PRC2 binding is increased genome-wide upon shPHF19L: from Figure 3C, it is clear that there is an increase at existing PRC2 peaks. But is there as well an increase in the number of PRC2 peaks? This needs to be analyzed by comparing peak calling in shPHF19L vs shCTRL, as well as a proper differential analysis of ChIP-seq signal in common peaks (using Limma or edgeR for example). This comment is actually valid for all ChIP-seq analyses of the paper.

3) While a CRISPR-Cas9 mediated knockout of PHF19L would be preferable to shRNA approach, at minimum, the authors should include a second shRNA construct. This would ensure that the interesting observations from this manuscript (enhanced PRC2 recruitment and or gene expression changes) are not an artefact of one shRNA knockdown. Targeted approaches (ChIP-qPCR and RT-qPCR) on few loci would be appropriate.

4) For Figure 2E, plots representing the detail of the data are needed, not average plots only. One could use scatterplots comparing PHF19 enrichment signal (log2 IP/INPUT) versus EZH2 (SUZ12, H3K27me3) enrichment. Such plots would give the reader insight on the co-occupancy of the factors. These plots should be complemented with a correlation score to assess the significance of this co-occupancy for example.

Suggested revisions:

1) Are any of these gene expression changes presented in Figure 5 directly related to loss of PHF19 or gain of MTF2/PRC2 binding? It is very important the authors include this data to show whether or not these expression changes are as a direct result of PHF19 knockdown. Rescue experiments would be recommended.

2) It would be interesting to mine the TCGA database. On top of the present analysis of PC3 and DU145 prostate cancer cell lines, it may be useful to document that PHF19L/S is also overexpressed in tumor versus normal cells in prostate cancers. Cell lines can sometimes diverge from what happens in tumors.

---

## [Author Response]

Essential revisions:1) Two important and unanimous critics relate to the overall lack of information about replicates and insufficient statistical assessments.Information on replication should be more comprehensive and systematically provided in Figure legends. For example (and not the least important), it is not known as to whether ChIP-seq experiments were performed in replicates or not. The use of spike-in is appreciated, but if there is no biological replicate, this should be acknowledged by the authors as a clear disclaimer in the text. This is particularly important as the conclusion from the ChIP-seq is at the center of the main conclusion of the study and also, because the increase in H3K27me3 (Figure 3A, right) is minor and would only be convincing if observed across replicates. On this point, the authors state that the increase is small albeit "significant", but no statistical test is presented.

We agree with the reviewers regarding the importance of using replicates in ChIP-seq experiments. To address this issue, we have performed a new set of ChIP-seq biological replicates for PHF19, H3K27me3, EZH2, SUZ12, and MTF2 in both conditions (shPHF19L and shCTR). Our initial conclusions were fully reproduced by these new ChIP-seq experiments, strengthening our conclusions. Moreover, using replicates allowed us to perform differential peak analysis supported by statistical tests (see next point). Notably, we confirmed a significant increase of H3K27me3 levels and of EZH2, SUZ12 and MTF2 recruitment, following knockdown of PHF19.

We have included this information on the replicates in the main text as well as in the corresponding figure legends. In several cases, the results using the first replicates are now shown in main figures (corresponding to Figure 2 and Figure 3 in the revised version of our manuscript), and that of the replicates, in the corresponding supplementary figures (Figure 2—figure supplement 1, and Figure 3—figure supplement 1). The differential peak analyses using all replicates were also mentioned in the text.

With respect to changes in H3K27me3, we have now clearly stated (both in the main text and in the figure legends) that H3K27me3 levels increased significantly (with a P-value of ≤ 10^-16^) in both replicates (Figure 3A and Figure 3—figure supplement 1).

Raw data and processed data for the new set of ChIP-seq experiments have been deposited in our GEO entry.

Indeed, there is generally poor statistical back-up of the results. Some p-values are provided for Figure 4B and Figure 6C, but most of the other comparisons lack statistical testing. For example, in Figure 4D, no statistical testing is done to evaluate the significance of the overlaps. Please provide appropriate statistical tests. Moreover, it is suggested that all p-values should be corrected for multiple testing – Benjamini-Hochberg procedure for example.

As suggested, we have now calculated statistical values and included this information in each figure of the revised version of our manuscript. We have also specified the test used for each case. In particular, to assess the statistical significance of all overlap between the sets shown in Venn diagrams (such as Figure 4D, mentioned by the reviewers), we tested three different methods: (i) Fisher’s exact test, (ii) a test based on the hypergeometric distribution, and (iii) GeneOverlap R package. As we observed similar results in all cases, we decided to include the P-values calculated using GeneOverlap R package (based in the Fisher's exact test) for all the Venn diagram overlaps included in our work.

Additionally, the introduction of a second ChIP-seq replicate (see response above) allowed us to perform differential peak analysis with the DiffBind R package. Using this algorithm, we have now identified a robust collection of peaks that significantly gain (or lose) ChIP-seq signals between the experimental and control conditions (see below, point 2).

2) About the conclusion that PRC2 binding is increased genome-wide upon shPHF19L: from Figure 3C, it is clear that there is an increase at existing PRC2 peaks. But is there as well an increase in the number of PRC2 peaks? This needs to be analyzed by comparing peak calling in shPHF19L vs shCTRL, as well as a proper differential analysis of ChIP-seq signal in common peaks (using Limma or edgeR for example). This comment is actually valid for all ChIP-seq analyses of the paper.

As requested, we now report the increase in the number of PRC2 peaks and target genes in the PHF19 knockdown cells. We have included this information in the new Figure 3B (H3K27me3) and Figure 3—figure supplement 1 (EZH2 and SUZ12). Moreover, we show that there is a significant increase in ChIP signal in both common and de novo targets (see boxplots below the Venn diagrams, Figure 3B and Figure 3—figure supplement 1). We have also included a new Supplementary file 1 with the total number of ChIP peaks and target genes (in control and knockdown conditions), and the corresponding lists of the target genes for each ChIP experiment included in our manuscript.

We thank the reviewers for this interesting suggestion about performing differential analysis. As mentioned above, the second set of ChIP-seq experiments allowed us to run the DiffBind R package, and to identify the peaks in which the change of signal was significant between shPHF19L and shCTR conditions. In all cases (PRC2 subunits and H3K27me3), DiffBind reported that the detected differential peaks were associated to a gain of signal in PHF19 knockdown (new Figure 3 and Figure 4—figure supplement 1), with a significant P value and FDR calculated by DiffBind (based on DESeq2 R package, similarly to edgeR).

3) While a CRISPR-Cas9 mediated knockout of PHF19L would be preferable to shRNA approach, at minimum, the authors should include a second shRNA construct. This would ensure that the interesting observations from this manuscript (enhanced PRC2 recruitment and or gene expression changes) are not an artefact of one shRNA knockdown. Targeted approaches (ChIP-qPCR and RT-qPCR) on few loci would be appropriate.

Following the suggestion of the reviewers, we have now used a second shRNA construct targeting the 3’ UTR of the *PHF19* gene (shPHF19L#B). We performed ChIP-qPCRs and RT-qPCRs on representative genes to validate the main observations of the manuscript. Our initial conclusions were fully reproduced when using this second shRNA. Moreover, we confirmed the specificity of the detected PHF19 ChIP peaks (new Figure 2—figure supplement 1) and the increased signals of H3K27me3, EZH2 (new Figure 3—figure supplement 1), and MTF2 (new Figure 4—figure supplement 1) in PHF19-depleted cells. Finally, we have also validated the changes in gene expression following PHF19 knockdown (new Figure 5).

4) For Figure 2E, plots representing the detail of the data are needed, not average plots only. One could use scatterplots comparing PHF19 enrichment signal (log2 IP/INPUT) versus EZH2 (SUZ12, H3K27me3) enrichment. Such plots would give the reader insight on the co-occupancy of the factors. These plots should be complemented with a correlation score to assess the significance of this co-occupancy for example.

As requested, we have performed the scatterplots comparing ChIP enrichment (now included in the new Figure 2E). In all cases, the correlation between PHF19 *vs*. the second ChIP-seq sample was high and statistically significant (values are provided in the corresponding figure legend). Importantly, we obtained similar results in our new set of ChIP-seq replicates (shown in the new Figure 2—figure supplement 1). Finally, we complemented our previous “average plots” by showing the distribution of values in a boxplot next to each metaplot (see new Figure 2E and Figure 2—figure supplement 1).

Suggested revisions:1) Are any of these gene expression changes presented in Figure 5 directly related to loss of PHF19 or gain of MTF2/PRC2 binding? It is very important the authors include this data to show whether or not these expression changes are as a direct result of PHF19 knockdown. Rescue experiments would be recommended.

To examine the direct implication of PHF19 in the effects we report here, we performed rescue experiments as requested by the reviewers. For this, we ectopically expressed FLAG-tagged PHF19 in PHF19-depleted or control cells. To knock down PHF19, we used an shRNA targeting the 3’ UTR of the gene (shPHF19L#B), such that only the endogenous *PHF19* was affected. ChIP-qPCR experiments on a set of representative genes demonstrated that MTF2 recruitment displayed the opposite behavior to PHF19: it increased in the absence of PHF19 but decreased to basal levels after PHF19 rescue. Moreover, changes in gene expression observed in the knockdown cells were reversed after overexpression of exogenous PHF19L. Our results clearly indicated that the effects reported herein directly depend on the presence of PHF19L. These results are presented in the new Figures 4 and 5.

2) It would be interesting to mine the TCGA database. On top of the present analysis of PC3 and DU145 prostate cancer cell lines, it may be useful to document that PHF19L/S is also overexpressed in tumor versus normal cells in prostate cancers. Cell lines can sometimes diverge from what happens in tumors.

As suggested, we have explored several databases (e.g., TCGA and ICGC) to investigate the expression of PHF19 in human prostate tumors. However, the results that were extracted from mining were inconclusive. High variability of PHF19 expression levels was detected between samples, which precludes us from determining any solid difference between normal and cancer prostate tissue. This is likely related to the fact that PHF19 itself, or variations of PRC2 complexes, are involved in many different biological processes, often related to cancer progression (see also PMID:30341152). Indeed, as reported in our manuscript, PHF19L positively regulates cell proliferation, but its downregulation is linked to invasion and angiogenesis.